# AMICABLE PERTURBATIONS

## ABSTRACT

Machine learning based classifiers have achieved incredible success in a variety of sectors such as college admissions, hiring and banking. However their ability to make classifications has not been fully exploited to understand how to improve undesirable classifications. We propose a new framework for finding the most efficient changes that could be made in the real world to achieve a more favorable classification, and term these changes *amicable perturbations*. We present a principled methodology for creating amicable perturbations and demonstrate their effectiveness on data sets from a variety of fields. Amicable perturbations differ from counterfactuals in that they are better suited to balance the effort-reward trade-off and lead to the most efficient plan of action. Unlike adversarial examples, which fool a classifier into making false prediction, amicable perturbations are intended to affect the true class of the data point. To this end, we develop a novel method for verifying that amicable perturbations change the true class probabilities. We also compare our results to those achieved by previous methods such as counterfactuals and adversarial attacks.

## 1 INTRODUCTION

The astounding accuracy of modern machine learning (ML) has lead to its widespread adoption in an increasing number of fields such as credit lending (Leo et al., 2019), college admissions (Martinez Neda et al., 2021), and healthcare (Sauer et al., 2022). When one of these classifiers returns an undesirable classification, however, it can leave users feeling unsatisfied and powerless. We propose a new framework to exploit the insight gained from these classifiers to efficiently change an undesirable classification. Consider a classifier that determines the credit risk associated with a loan application that a bank could use to determine to whom they should offer a loan. These classifiers' understanding of what makes an individual a good or bad credit risk could also be used to find the changes an individual should make to improve their credit worthiness, after their application is rejected. The useful actions may be obvious: pay down existing debts, increasing income and money in saving, etc., but an amicable perturbation could suggest the precise combination of these changes that would reach a specific goal (such as less than 10% chance of default) most efficiently, for example changes that can be made in the least amount of time. We call these classifier informed changes *amicable perturbations*. We present a few more illustrative applications of amicable perturbations: (a) If a classifier gives low odds of survival to a patient in an understaffed hospital, then an amicable perturbation could suggest the course of treatment that would double the patients odds of survival while requiring the least staff hours. (b) When an ML classifier decides not to pass a job applicant's resume on to an employer, an amicable perturbation could suggest the skills the applicant could acquire in the least amount of time that would lead to a high likelihood of receiving an interview request. (c) If an ML classifier has been trained to predict at which price range a product will sell, an amicable perturbation could suggest the cheapest modifications to a product that would bring it into a more premium price range and enhance marketability.

**Overview of Amicable Perturbations and Contributions**. Our framework consist of a) a concept of real world feasibility of a change through the notion of an actionable set (i.e., $\tilde{\mathbf{x}}_{AP} \in \mathcal{A}(\mathbf{x})$); b) a measure of the effort/cost required to make a change (denoted by $d_{\mathcal{X}}(\mathbf{x}, \tilde{\mathbf{x}}_{AP})$); c) the goal of the change, denoted by a target set of probability distributions $T$; and d) a measure of how close the *true class probabilities* $\tilde{\mathbf{y}}$ of $\tilde{\mathbf{x}}_{AP}$ comes to achieving the desired goal through the notion of statistical distance to the target set, $d_{\mathcal{Y}}(\tilde{\mathbf{y}}, T)$. Generating amicable perturbations can then be formulated as minimizing a weighted sum of the statistical distance to the target set and the effort/cost measure,

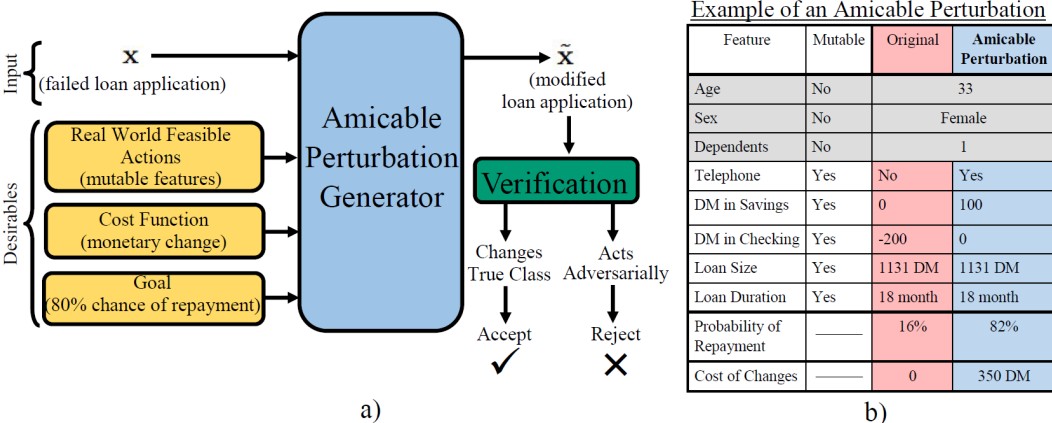

Figure 1: a) Illustration of the framework for creating amicable perturbations. b) Example of the original input and corresponding Amicable Perturbation on German Credit data set.

|  | Amicable Perturbations (AP) | Counterfactuals (CF) | Adversarial Attacks (AA) |
|---|:---:|:---:|:---:|
| Intended to give advice | ✓ | ✓ | ✗ |
| Verifies real world efficacy | ✓ | ✗ | ✗ |
| Flexible definition of final goal | ✓ | ✗ | ✗ |
| Designed for optimal efficiency | ✓ | ✗ | ✓ |

Table 1: Comparison of amicable perturbations to counterfactuals and adversarial attacks.

subject to the constraint that the change $\tilde{\mathbf{x}}_{AP}$ is actionable. To ensure that an amicable perturbation does not behave like an adversarial attack (affecting the classifier's decision but not the true class probabilities) we propose a novel *verification procedure*. We take advantage of the fact that we have access to the original data point and the modified point (never the case in adversarial settings). This allows us to use a verifier function $V$ that takes in two data points simultaneously. We use the value of $V(\mathbf{x}, \tilde{\mathbf{x}}_{AP})$ to determine whether the changes are likely to produce a truly different outcome. We demonstrate the effectiveness of this procedure on data sets from a variety of fields.

**Distinction from Adversarial Examples & Counterfactuals** Previous work on modifying an input $\mathbf{x}$ to change its classification can be grouped into two main categories. 1) *Adversarial attacks* $\tilde{\mathbf{x}}_{AA}$ are small perturbations which also seek to change the classification of an input, but they are intended to "fool" the classifier without changing the true class of the data point (Szegedy et al., 2013; Goodfellow et al., 2015; Madry et al., 2019; Carlini & Wagner, 2017b). 2) A *Counterfactual* for an input $\mathbf{x}$ is a similar input $\tilde{\mathbf{x}}_{CF}$ which leads to a different classification. Counterfactuals were originally proposed as a way to explain the decisions of a black-box classifier (Wachter et al., 2017; Dhurandhar et al., 2018; Guidotti et al., 2018; Van Looveren & Klaise, 2021). Many works have repurposed counterfactuals to also give advice on changing an unfavorable outcome (Mothilal et al., 2020; Ustun et al., 2019; Poyiadzi et al., 2020; Karimi et al., 2021) sometimes using different (but highly related) terms such as recourse or inverse classification. In contrast to adversarial attacks and counterfactuals, amicable perturbations $\tilde{\mathbf{x}}_{AP}$ have the express purpose of changing real world outcomes (true class probabilities). Consider an individual turned down for a loan: a counterfactual's purpose is to get the individual approved by the classifier, but an amicable perturbation's purpose is to help the individual improve their actual odds of paying off the loan. This is significant because counterfactuals are created with similar methods as adversarial attacks (Pawelczyk et al., 2022), which are known to "fool" classifiers. Our objective is shared by König et al. (2023), but we also introduce flexible goals and a cost reward-framework more conducive to this task. While König et al. (2023) ensure trustworthiness against non-causal correlations through the use of Structural Causal Models (SCMs), we provide trustworthiness against classifier weakness through the introduction of a verification procedure. For more information on adversarial attacks and counterfactuals we direct the reader to surveys Akhtar & Mian (2018) and Guidotti (2022). Table 1 highlights key distinctions and similarities between amicable perturbations, counterfactuals and adversarial attacks.

## 2 AMICABLE PERTURBATIONS

**Problem Setting and Goals:** Suppose there is an unknown distribution $(\mathbf{x}, C) \sim \mathcal{D}$. Here $\mathbf{x}$ is a member of the input space $\mathcal{X} \subset \mathbb{R}^m$ and $C \in \{1, ..., k\}$ is the class of $\mathbf{x}$. We define the class probabilities $\mathbf{y} := (\mathbb{P}(C = 1|\mathbf{x}), ..., \mathbb{P}(C = k|\mathbf{x}))$. We let $\mathcal{Y}$ denote the $k$-simplex and use a classifier $M : \mathcal{X} \to \mathcal{Y}$ to estimate the $\mathbf{y}$ corresponding to an input $\mathbf{x}$. Let $\tilde{C}$ and $\tilde{\mathbf{y}}$ be the class and class probabilities of $\tilde{\mathbf{x}}$. Our goal in designing an amicable perturbation is: *Given an input $\mathbf{x}$ with an undesirable classification $M(\mathbf{x})$, find the most efficient real world actions to create a modified input $\tilde{\mathbf{x}}$ such that the corresponding true probabilities $\tilde{\mathbf{y}}$ (and not just $M(\tilde{\mathbf{x}})$) are more desirable.*

**Real World Actionability:** Amicable perturbations should only suggest modifications that are feasible in the real world (e.g., not decreasing an individual's age). To this end, we introduce: *the Actionable Set $\mathcal{A}(\mathbf{x})$ of a data point $\mathbf{x}$ is the set of all perturbations of $\mathbf{x}$ that are feasible in the real world.* For example, if $\mathcal{X}$ represents loan applications with $x_1$ the age of the applicant, $x_2$ the applicant's credit score, $x_3$ the amount of credit and $x_4$ the loan duration, and the applicant must obtain a loan quickly we would use $\mathcal{A}(\mathbf{x}) = \{\tilde{\mathbf{x}} \in \mathcal{X} | \tilde{x}_1 = x_1, \tilde{x}_2 = x_2\}$, i.e. the applicant can change the size and duration of the loan they request, but not their age or credit score. Alternatively, if the applicant can wait for a longer time to get the loan, we could define the actionable set as $\mathcal{A}(\mathbf{x}) = \{\tilde{\mathbf{x}} \in \mathcal{X} | \tilde{x}_1 \geq x_1\}$, i.e. allowing flexibility to change the credit score over time. We note that the actionable set is dependent on both the specific data point, as well as the underlying context.

**Efficiency:** The definition of the most efficient change depends on the context of the problem and could involve a well defined value such as "cost in dollars" or more nebulous value such as "amount of effort required." We characterize this value with a function $d_{\mathcal{X}} : \mathcal{X} \times \mathcal{X} \to \mathbb{R}$, where $d_{\mathcal{X}}(\mathbf{x}, \tilde{\mathbf{x}})$ is the cost of changing $\mathbf{x}$ to $\tilde{\mathbf{x}}$. For example, if $\mathbf{x}$ represents a resume and $\tilde{\mathbf{x}}$ is a modified resume, then $d_{\mathcal{X}}(\mathbf{x}, \tilde{\mathbf{x}})$ could represent the time it would take to acquire the attributes listed on resume $\tilde{\mathbf{x}}$, but not on $\mathbf{x}$. We note this function may not be a true distance measure. For example, if $d_{\mathcal{X}}$ represents the difference in financial cost between two courses of medical treatment, then $d_{\mathcal{X}}(\mathbf{x}, \tilde{\mathbf{x}})$ should be negative when $\tilde{\mathbf{x}}$ is more affordable than $\mathbf{x}$. This function (and the actionable set) may need to be designed with the help of a subject matter expert.

**Desirability:** We now define what we mean by a desirable outcome—the goal of an amicable perturbation. *The Target Set $T$ is the set of all elements of $\mathcal{Y}$ that would be an acceptable result of an amicable perturbation.* If we wish to belong to a desirable class $w$ with probability no less than $p$, the target set would have the form $T = \{\mathbf{z} \in \mathcal{Y} | z_w \geq p\}$. If our goal is rather to avoid some undesirable class $u$, $T$ could be of the form $T = \{\mathbf{z} \in \mathcal{Y} | z_u \leq q\}$ for a fixed $q$. More generally, suppose we wish to belong to a set of desirable classes $\mathcal{W}$ with probability at least $p$ and we wish to belong to a set of undesirable classes $\mathcal{U}$ with probability no greater than $q$. Then $T$ may be written

$$T = \left\{ \mathbf{z} \in \mathcal{Y} \,\middle|\, \sum_{i \in \mathcal{W}} z_i \geq p, \sum_{i \in \mathcal{U}} z_i \leq q \right\}. \tag{1}$$

We next quantify how close an amicable perturbation $\tilde{\mathbf{x}}$ comes to its goal in a principled manner. To do this, we first choose a measure of statistical distance $D(\tilde{\mathbf{y}} || \mathbf{z})$ (such as Kullback-Leibler (KL) Divergence). We then denote $d_{\mathcal{Y}}(\tilde{\mathbf{y}}, T)$ as the distance of $\tilde{\mathbf{y}}$ to the target set $T$, defined as follows:

$$d_{\mathcal{Y}}(\tilde{\mathbf{y}}, T) = \min_{\mathbf{z} \in T} D(\tilde{\mathbf{y}} || \mathbf{z}). \tag{2}$$

We next formally define amicable perturbations. Let $\epsilon$ represent budget —the amount of work we are willing to perform, and $\delta$ represent tolerance —how close the final result is to our target set $T$.

**Definition 1 (($\epsilon, \delta$)-Amicable Perturbation)** *$\tilde{\mathbf{x}}$ is an ($\epsilon, \delta$)-amicable perturbation for $\mathbf{x}$ and $T$ if*

    *1. $d_{\mathcal{X}}(\mathbf{x}, \tilde{\mathbf{x}}) \leq \epsilon$*

    *2. $d_{\mathcal{Y}}(\tilde{\mathbf{y}}, T) \leq \delta$*

    *3. $\tilde{\mathbf{x}} \in \mathcal{A}(\mathbf{x})$.*

**Real-world Verifiability of Amicable Perturbations**: Note that amicable perturbations are defined with respect to the true class probabilities $\tilde{\mathbf{y}}$ because amicable perturbations should have an effect in the real world. Notwithstanding, $\tilde{\mathbf{y}}$ is unknown and we must use $M(\tilde{\mathbf{x}})$ to create our amicable

perturbations (more details in Section 3), which introduces the risk that we might produce an $\tilde{\mathbf{x}}$ that has the desired effect on $M(\tilde{\mathbf{x}})$ but not $\tilde{\mathbf{y}}$ (like an adversarial example). Ilyas et al. (2019) suggest that this occurs because $M$ learns correlations that are found in the training data, $\{\mathbf{x}^{(j)}, C^{(j)}\}_{j=1}^{n}$, but do not generalize outside the training data. Verifying $\tilde{\mathbf{x}}$ is similar to detecting adversarial examples which has been the object of significant research (Yang et al., 2020; Roth et al., 2019; Fidel et al., 2020; Carlini & Wagner, 2017a) with no satisfactory solution. Fortunately, we have an important advantage over detecting adversarial examples: *we know the original data point* $\mathbf{x}$ *and exactly how it was modified, i.e.,* $\tilde{\mathbf{x}}$. To capitalize on this knowledge, we propose a novel verification procedure using a classifier $V : \mathcal{X} \times \mathcal{X} \to [0, 1]$ which compares two inputs simultaneously and predicts the probability of the inputs belonging to the same class: the value of $V(\mathbf{x}, \tilde{\mathbf{x}})$ estimates $\mathbb{P}(C = \tilde{C} | \mathbf{x}, \tilde{\mathbf{x}})$. We create $V$ using the same structure as $M$ (only changing the dimension of the input and output) and training on the new set of *difference training data* $\{(\mathbf{x}^{(i)}, \mathbf{x}^{(j)}), z^{(i,j)}\}_{1 \leq i,j \leq n}$, where $z^{(i,j)} = \mathbb{1}[C^{(i)} = C^{(j)}]$. Because $V$ is trained on a fundamentally different classification problem than $M$ it will learn different features and will not be affected by the same non-generalizable correlations as $M$. We can also estimate $\mathbb{P}(C = \tilde{C} | \mathbf{x}, \tilde{\mathbf{x}})$ using $M$ by calculating $\sum_{i=1}^{k} M_i(\mathbf{x}) M_i(\tilde{\mathbf{x}})$. If $\tilde{\mathbf{x}}$ acts adversarially we would expect $\sum_{i=1}^{k} M_i(\mathbf{x}) M_i(\tilde{\mathbf{x}})$ to be very small while $V(\mathbf{x}, \tilde{\mathbf{x}})$ is large. If $\tilde{\mathbf{x}}$ is not adversarial we would expect similar values from both $\sum_{i=1}^{k} M_i(\mathbf{x}) M_i(\tilde{\mathbf{x}})$ and $V(\mathbf{x}, \tilde{\mathbf{x}})$. Accordingly we use the discrepancy between these two estimates:

$$\Delta(\mathbf{x}, \tilde{\mathbf{x}}) = \left| V(\mathbf{x}, \tilde{\mathbf{x}}) - \sum_{i=1}^{k} M_i(\mathbf{x}) M_i(\tilde{\mathbf{x}}) \right| \tag{3}$$

to verify that an amicable perturbation $\tilde{\mathbf{x}}$ can be trusted to have the anticipated result in the real world. Specifically, if $\Delta(\mathbf{x}, \tilde{\mathbf{x}}) < \gamma$, then we accept $\tilde{\mathbf{x}}$ as a verified amicable perturbation; otherwise, we reject. In Section 3, we describe how to select the threshold $\gamma$.

**Relation to Previous Work:** Previous work on modifying an input to change its classification has focused on counterfactuals and adversarial attacks. A counterfactual $\tilde{\mathbf{x}}_{CF}$ to an input $\mathbf{x}$ is a similar input that results in a different classification. Wachter et al. (2017) suggested using counterfactuals to explain the logic an ML classifier, and subsequent works (Mothilal et al., 2020; Ustun et al., 2019; Poyiadzi et al., 2020; Karimi et al., 2021) suggested making counterfactuals *actionable* by only allowing changes that could be made in the real world, that is $\tilde{\mathbf{x}}_{CF} \in \mathcal{A}(\mathbf{x})$. Counterfactuals are well suited to explaining a classifiers decisions, but amicable perturbations have several key advantages for providing advice to change the real world. First, amicable perturbations are more precise and flexible in how their goals are defined. In contrast, counterfactuals focus only on the final classification, and a change that leads to only a 51% chance of the desired result is a valid counterfactual. Secondly, counterfactuals typically minimize $||\mathbf{x} - \tilde{\mathbf{x}}_{CF}||$ for an $\ell_p$ norm which fails represent the real world costs of a change. Alternatively, amicable perturbations minimize real world cost which (in conjunction with the principled measure of distance to a target) leads to optimal advice (see examples in Section 4). Finally, amicable perturbation's verification procedure helps ensure that changes will have the intended effect on the real world.

Verification is essential because of the existence of adversarial attacks (Akhtar & Mian, 2018): algorithms which create inputs $\tilde{\mathbf{x}}_{AA}$ (called adversarial examples) that lead to misclassifications. Counterfactuals and adversarial examples are both created by solving the some version of the problem

$$\tilde{\mathbf{x}}_{CF}, \tilde{\mathbf{x}}_{AA} = \arg \min_{\tilde{\mathbf{x}}} \ loss(\tilde{\mathbf{x}}, w) + \lambda ||\tilde{\mathbf{x}}, \mathbf{x}|| \tag{4}$$

where $w$ is the desired class, $\lambda$ is a tuning parameter, $|| \cdot ||$ is a norm and a variety of $loss$ functions could be used. It is concerning that counterfactuals (which are often assumed to change the actual class of and input), and adversarial examples (which fool a classifier) are created using such similar methods. In fact, Pawelczyk et al. (2022) showed that many of the most common adversarial attacks and counterfactual generation algorithms solve nearly identical versions of equation 4 and produce very similar results. Wachter et al. (2017) noted this concern when introducing counterfactuals, but they dismissed it because the adversarial attacks of the time 1) modified many more features than counterfactuals and 2) were targeted almost exclusively at image data whereas counterfactuals were proposed for use on tabular data. Since that time, Su et al. (2019) demonstrated that adversarial attacks can be effective when changing a very small number of features (just one pixel), and several works (Ballet et al., 2019; Mathov et al., 2020; Cartella et al., 2021; Kumar et al., 2021) have shown that adversarial examples exist even on tabular data sets. This implies verification is necessary to achieve a result that can be trusted to change the true class probabilities.

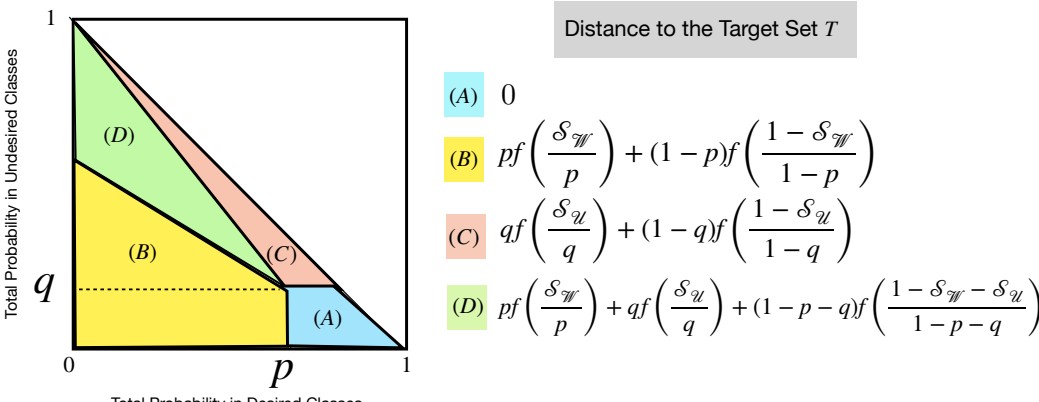

Figure 2: Illustration of the statistical distance based cost $d_{\mathcal{Y}}(\tilde{\mathbf{y}}, T)$ derived in Theorem 1 as a function of the total probability in desired vs. undesired classes. Although the cost function takes different functional form(s) in the four regions, it is continuously differentiable in the entire space.

## 3   GENERATING AMICABLE PERTURBATIONS

**Two Step Creation Method:** We now present and discuss the general optimization framework for creating amicable perturbations. Ideally, we would like to solve the following optimization problem: $\arg\min_{\tilde{\mathbf{x}} \in \mathcal{A}(\mathbf{x})} d_{\mathcal{Y}}(\tilde{\mathbf{y}}, T) + \lambda d_{\mathcal{X}}(\tilde{\mathbf{x}}, \mathbf{x})$, where the scalar parameter $\lambda$ balances the effort($\epsilon$)-reward($\delta$) trade-off. Although solving this optimization would lead to a guaranteed and effective amicable perturbation, the problem is that $\tilde{\mathbf{y}}$ is unknown. To deal with this challenge, we instead propose the following two-step procedure where: in Step 1, we treat $M(\tilde{\mathbf{x}})$ as a surrogate for $\tilde{\mathbf{y}}$, and in Step 2, we use a verification algorithm to ensure that $\tilde{\mathbf{x}}$ is not just fooling the classifier.

$$\underbrace{\arg\min_{\tilde{\mathbf{x}} \in \mathcal{A}(\mathbf{x})} d_{\mathcal{Y}}(M(\tilde{\mathbf{x}}), T) + \lambda d_{\mathcal{X}}(\tilde{\mathbf{x}}, \mathbf{x})}_{\text{Step 1}} \rightarrow \underbrace{\text{Verify } M(\tilde{\mathbf{x}}) \approx \tilde{\mathbf{y}}}_{\text{Step 2}} \rightarrow \text{Amicable Perturbation} \quad (5)$$

**Properties of Statistical Distance $d_{\mathcal{Y}}$ to the Target:**   Understanding $d_{\mathcal{Y}}$ will be key to completing Step 1. In Section 2 we defined: $d_{\mathcal{Y}}(\tilde{\mathbf{y}}, T) = \min_{\mathbf{z} \in T} D(\tilde{\mathbf{y}}\|\mathbf{z})$; Here we will analyze this function and show that it has properties conducive to solving the optimization in Step 1. For our analysis, we set $D(\tilde{\mathbf{y}}\|\mathbf{z})$ to be an $f$-divergence: a broad class of measures including KL-divergence, total-variation (TV) and other commonly used statistical distances. An $f$-divergence is defined as $D(\tilde{\mathbf{y}}\|\mathbf{z}) = \sum_{i=1}^{k} \mathbf{z}_i f\left(\frac{\tilde{\mathbf{y}}_i}{\mathbf{z}_i}\right)$, where $f$ is a convex function satisfying $f(1) = 0$ and $f(0) = \lim_{x \to 0^+} f(x)$ (Polyanskiy & Wu, 2022). In Theorem 1 we find that $d_{\mathcal{Y}}$ has an easy to use differentiable closed form as long as $f$ is twice differentiable (for instance, KL Divergence) and the target set $T$ has the very general form (1).

**Theorem 1** *If $D(\tilde{\mathbf{y}}\|\mathbf{z})$ is an $f$-Divergence with twice differentiable $f$ and $T$ is of form (1), then*

$$d_{\mathcal{Y}}(\tilde{\mathbf{y}}, T) = \begin{cases} 0 & \text{if } \mathcal{S}_{\mathcal{W}} \geq p \text{ and } \mathcal{S}_{\mathcal{U}} \leq q \\ pf\left(\frac{\mathcal{S}_{\mathcal{W}}}{p}\right) + (1-p)f\left(\frac{1-\mathcal{S}_{\mathcal{W}}}{1-p}\right) & \text{if } \mathcal{S}_{\mathcal{W}} < p \text{ and } \mathcal{S}_{\mathcal{U}} \leq (1-\mathcal{S}_{\mathcal{W}})\left(\frac{q}{1-p}\right) \\ qf\left(\frac{\mathcal{S}_{\mathcal{U}}}{q}\right) + (1-q)f\left(\frac{1-\mathcal{S}_{\mathcal{U}}}{1-q}\right) & \text{if } \mathcal{S}_{\mathcal{U}} > q \text{ and } \mathcal{S}_{\mathcal{W}} \geq (1-\mathcal{S}_{\mathcal{U}})\left(\frac{p}{1-q}\right) \\ pf\left(\frac{\mathcal{S}_{\mathcal{W}}}{p}\right) + qf\left(\frac{\mathcal{S}_{\mathcal{U}}}{q}\right) & \text{if } \mathcal{S}_{\mathcal{U}} > (1-\mathcal{S}_{\mathcal{W}})\left(\frac{q}{1-p}\right) \\ \quad + (1-p-q)f\left(\frac{1-\mathcal{S}_{\mathcal{W}}-\mathcal{S}_{\mathcal{U}}}{1-p-q}\right) & \text{and } \mathcal{S}_{\mathcal{W}} < (1-\mathcal{S}_{\mathcal{U}})\left(\frac{p}{1-q}\right) \end{cases}, \quad (6)$$

*where $\mathcal{S}_{\mathcal{W}} = \sum_{i \in \mathcal{W}} \tilde{y}_i$ and $\mathcal{S}_{\mathcal{U}} = \sum_{i \in \mathcal{U}} \tilde{y}_i$. Furthermore, $d_{\mathcal{Y}}(\tilde{\mathbf{y}}, T)$ is continuously differentiable.*

Theorem 1 shows that $d_{\mathcal{Y}}(\tilde{\mathbf{y}}, T)$ takes on a piece-wise form, and Figure 3 illustrates how the pieces of (6) divide the probability space. Despite its piece-wise form, this function is continuously differentiable over its entire domain, which will be significant when solving the optimization problem in Step 1. The proof of Theorem 1 and additional results about $d_{\mathcal{Y}}$ may be found in the Appendix.

---

**Algorithm 1** Generating Amicable Perturbations

---

**Require:** Classifiers $M$ & $V$, point $\mathbf{x}$, target family $T$, learning rate $\alpha$, verification-cut off $\gamma$

    $\tilde{\mathbf{x}} \leftarrow \mathbf{x}$
    **while** $\tilde{\mathbf{x}}$ not converged **do**
        $\mathbf{g} \leftarrow \nabla_{\tilde{\mathbf{x}}} \left( d_{\mathcal{Y}}(M(\tilde{\mathbf{x}}), T) + \lambda d_{\mathcal{X}}(\tilde{\mathbf{x}}, \mathbf{x}) + b(\tilde{\mathbf{x}}) + p(\tilde{\mathbf{x}}) \right)$
        $\mathbf{g}_j \leftarrow 0$ for all immutable features $j$.
        $\tilde{\mathbf{x}} \leftarrow \tilde{\mathbf{x}} - \alpha g$
    **end while**
    $\tilde{\mathbf{x}} = cond(\tilde{\mathbf{x}})$ (project onto the coherent space)
    $\epsilon, \delta = d_{\mathcal{X}}(\tilde{\mathbf{x}}, \mathbf{x}), d_{\mathcal{Y}}(M(\tilde{\mathbf{x}}), T)$
    **if** $\epsilon$ and $\delta$ requirements NOT met **then**
        Adjust $\lambda$ (see text for explanation)
        Return to while loop
    **end if**
    **if** $\left| V(\mathbf{x}, \tilde{\mathbf{x}}) - \sum_{i=1}^{k} M_i(\mathbf{x}) M_i(\tilde{\mathbf{x}}) \right| \geq \gamma$ **then**
        Adjust problem parameters (see text for explanation)
        Restart algorithm
    **end if**
    return $\tilde{\mathbf{x}}$

---

**Solving Step 1:** We solve our optimization problem using gradient descent which requires us to use differentiable models $M$ and formulate $d_{\mathcal{X}}$ in a differentiable manner ($d_{\mathcal{Y}}$ is differentiable according to Theorem 1). We modify our gradient descent to address two challenges. (1) We must insure that our solution is actionable: $\tilde{\mathbf{x}} \in \mathcal{A}(\mathbf{x})$. (2) Our solution $\tilde{\mathbf{x}}$ must follow any formatting rules associated with the data set (Boolean variables must be either 0 or 1, categorical features must respect one-hot encoding, etc.). A perturbation that follows these formatting rules is called *coherent*. To solve these two difficulties, we first assume $\mathcal{A}(\mathbf{x}) = \{\tilde{\mathbf{x}} | l_i \leq \tilde{\mathbf{x}}_i \leq u_i, 1 \leq i \leq m\}$ for some set of lower bounds $\{l_i\}_{i=1}^{m}$ and upper bounds $\{u_i\}_{i=1}^{m}$. An attribute is *immutable* if $l_i = u_i$. We ensure actionability by setting all elements of the gradient corresponding to immutable features to zero and adding a large penalty $b(\tilde{\mathbf{x}})$ term to the objective function which punishes points for leaving the actionable set. To ensure coherence, we project the result of our gradient descent onto the coherent space by using a function $cond : \mathbb{R}^m \rightarrow \mathcal{X}$ which performs the appropriate value rounding to make an input coherent. We found it useful to introduce a second penalty term $p(\tilde{\mathbf{x}})$ which requires that any one-hot encoded features sum to 1. This ensures our answers never stray too far from a coherent point and improves robustness. Details on $b$, $p$ and $cond$ may be found in the Appendix. In practice we also found it useful to replace regular gradient descent with the ADAM algorithm (Kingma & Ba, 2014).

**Solving Step 2:** In Section 2, we discussed the necessity of verifying amicable perturbations and suggested that an amicable perturbation can be trusted if $\Delta(\mathbf{x}, \tilde{\mathbf{x}}) = \left| V(\mathbf{x}, \tilde{\mathbf{x}}) - \sum_{i=1}^{k} M_i(\mathbf{x}) M_i(\tilde{\mathbf{x}}) \right|$ is smaller than a threshold $\gamma$. To find the proper cut-off $\gamma$, we first decide on an acceptable risk of eliminating a truly effective amicable perturbation (we use 10%). To find the $\gamma$, corresponding to this risk, we calculate $\Delta(\mathbf{x}^{(i)}, \mathbf{x}^{(j)})$ for a sufficiently large number of pairs $(\mathbf{x}^{(i)}, \mathbf{x}^{(j)})$ from the testing data such that $C^{(i)} \neq C^{(j)}$. We can now pick $\gamma$ such that only the desired percentage of $\Delta(\mathbf{x}^{(i)}, \mathbf{x}^{(j)})$ values (e.g. 10%) are above $\gamma$. The verification procedure is now reduced to eliminating any amicable perturbation that results in $\Delta(\mathbf{x}, \tilde{\mathbf{x}}) > \gamma$.

**Adjusting for Suitability and Verifiability:** When creating amicable perturbations we will often have a particular budget ($\epsilon$) or tolerance ($\delta$) bound we need to satisfy. To find a suitable amicable perturbation we repeat Step 1 of our process adjusting $\lambda$ until the desired budget or tolerance is met: increasing $\lambda$ to decrease $\epsilon$ and decreasing $\lambda$ to decrease $\delta$. It may also be appropriate to use a variety of $\lambda$ values and plot the $\epsilon$ and $\delta$ values of each resulting amicable perturbation (see Section 4). The user may then select a perturbation they see as offering particularly good value. When an amicable perturbation fails the verification step, there are a few recourses. (1) Sometimes it is sufficient to decrease $\lambda$, putting greater emphasis on reaching the target set. (2) "Shrink" the target set (increase the value of $p$ and decrease the value of $q$) in order to force the algorithm to find more effective changes. (3) Add a random perturbation to $\mathbf{x}$ in order to move the starting point away from the adversarial example. We show the entire procedure in Algorithm 1.

| Data Set | Adult Income | Law School Success | Diabetes Prediction | German Credit |
|---|---|---|---|---|
| **Target Set:** $T$ | ≥90% of high income | ≥85% chance of passing | ≤25% chance of Diabetes | ≥80% chance of repayment |
| **Mutable Features** | Weekly hours worked, education level, job type, employer type | Law School Grades, location of BAR exam | BMI, smoking, fruit, vegetable and alcohol consumption, healthcare, education, income, physical activity | Loan size, loan duration, telephone, money in checking account, money in savings account |
| **Cost Measure** $d_{\mathcal{X}}$ | Combination of time to improve education and hours worked per week | Combination of effort to improve grades and physical distance to travel to take BAR | Effort required to loose weight, change health habits, education and income as weighted 2-norm | Total monetary difference in Deutsche Marks (DM) |
| **Immutable Features** | Age, sex, race, marital status | Sex, race, LSAT score, undergraduate GPA | Age, sex, blood pressure, cholesterol, stroke, heart disease, mental, physical and general health, difficulty walking | Age, sex, marital status income, dependents, loan purpose, employment length and type, housing, credit history, collateral |
| **Model Accuracy** | 80% | 77% | 75% | 75% |

Figure 3: Table containing details on data sets used for testing.

## 4 EXPERIMENTAL RESULTS

**Experimental Set Up:** We compare amicable perturbations, counterfactuals and adversarial attacks on four data sets from different fields; additional data set details in Figure 3 and Section 5.3.1.

*Adult Income* (Becker & Kohavi, 1996): This data set contains demographic information on Americans labelled by whether they had a high income. The actionable set $\mathcal{A}(\mathbf{x})$ allows individuals to increase their education, change jobs and adjust their weekly work hours. The cost function $d_{\mathcal{X}}$ sums the expected number of years to improve education, a one-year cost to change jobs and the square of the change in hours worked (weighted so 3 hours is equal to a year spent on education).
*Law School Success* (Wightman, 1998): This data set contains information on law school students labelled by whether they passed the BAR exam. In $\mathcal{A}(\mathbf{x})$ we allow changes to law school grades (through more studying) and the region where the exam is taken. The cost function $d_{\mathcal{X}}$ sums the increase in grades and the physical distance travelled to take the BAR. Moving to an adjacent region (e.g. Far West to North West) is weighted the same as increasing grades by one standard deviation.
*Diabetes Prediction* (for Disease Control & , CDC): The individuals in this data set are labelled by whether they have diabetes. We define $\mathcal{A}(\mathbf{x})$ to allow changes in health habits, BMI, education and income. We use a weighted 2-norm for $d_{\mathcal{X}}$ to represent the relative difficulty of making changes. For example, starting to get regular physical activity is weighted the same as dropping one BMI.
*German Credit* (Hofmann, 1994): This data set contains loan applications. In $\mathcal{A}(\mathbf{x})$, we allow for changes to the loan duration and size and funds in the checking and savings accounts. We use $d_{\mathcal{X}}$ to measure the total difference in Deutsche Marks (DM) over all elements of the application.

**Other Methods:** We compare our results against counterfactuals created using the original method proposed to create counterfactuals (Wachter et al., 2017) and the diverse counterfactuals (DICE) method in (Mothilal et al., 2020), the most cited methods in the literature. We also compare amicable perturbations against the Carlini & Wagner (2017b) $\ell_2$ adversarial attack, one of the most well known and effective adversarial attacks. The counterfactuals belong to the same actionable set as the amicable perturbations, but the adversarial examples need not be actionable or even coherent.

**Models:** Gradient boosted tree algorithms Friedman (2001) are considered state of the art architectures for tabular data classification Shwartz-Ziv & Armon (2021). Unfortunately, these models are not differentiable and cannot be used with our framework. Instead we use neural networks which we tuned until they provide accuracy on par with gradient boosted tree models on the same data set. Details on our models' structure and training may be found in the appendix.

**Representative Amicable Perturbations and Trade-off between cost/desirability:** We first examine two representative examples of how amicable perturbations behave differently than counterfactuals for specific individuals. Figure 4 shows a plot of the $\epsilon/\delta$ values of amicable perturbations and counterfactuals for one individual in the Law School data set and one individual in the Adult Income data set. We examine the results from the Law School data set: The amicable perturbation labelled $AP_1$ suggests only a mild (0.2 standard deviation) increase in grades and the relatively short move from the Far West to the Great Lakes region resulting in a small $11\%$ increase in the chance of passing the BAR. On the other hand, $AP_2$ suggest a larger increase in grades and a longer move

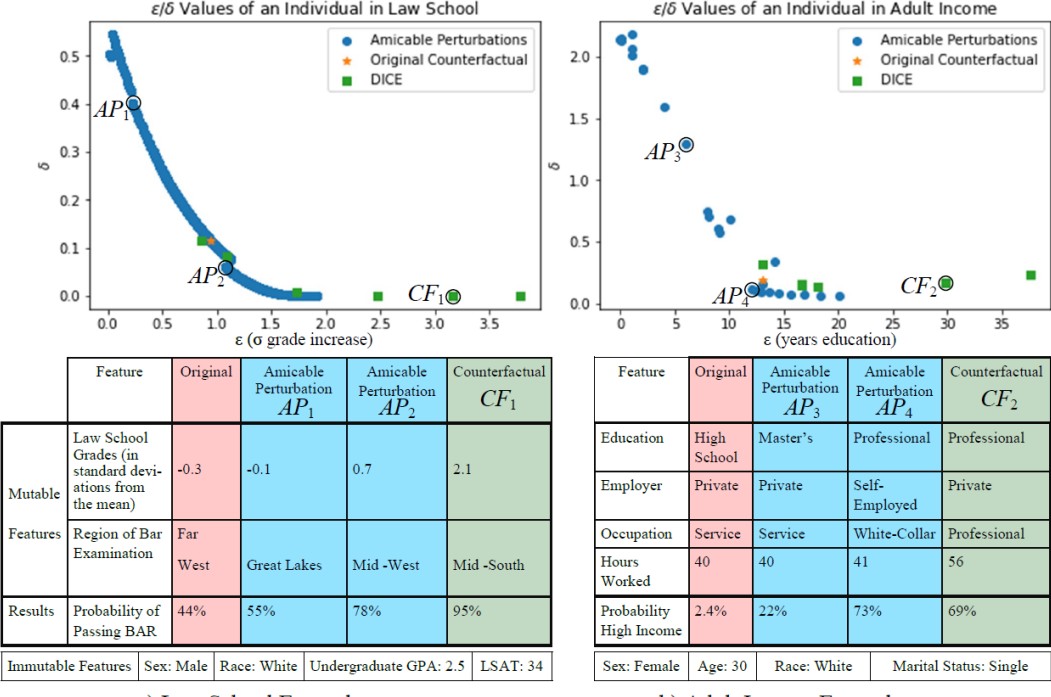

a) Law School Example        b) Adult Income Example

Figure 4: Cost-Benefit plots of amicable perturbations and counterfactuals for an individually from the Law School data set (a) and an individual in the Adult Income data set (b).

which results in a much larger $34\%$ increase to the odds of success. Finally the counterfactual $CF_1$ suggest an enormous increase in grades and massive cross country move to achieve $51\%$ increase in the odds of success. Turning our attention to the Adult Income example: $AP_3$ suggests a relatively simple increase in education to the masters level resulting in a $20\%$ increase to the odds of a high income. Alternatively, $AP_4$ achieves an $71\%$ increase by suggesting far more changes including a professional degree and becoming self-employed. The counterfactual $CF_2$ does not suggest becoming self-employed and produces a smaller $67\%$ increase in the odds of high income despite also suggesting a professional degree and a drastic 16 hour increase in the hours worked per week.

These examples illustrates two trends: 1) Amicable perturbations offer both low-cost/low-reward (large-$\delta$/small-$\epsilon$) and high-cost/high-reward options, whereas counterfactual methods Wachter et al. (2017); Mothilal et al. (2020) only offer high-cost options. This is because amicable perturbations are defined by distance to the target set, but counterfactuals are defined as belonging to the desirable class. That rules out any advice that doesn't result in the desirable class being the most likely class. 2) Counterfactuals are prone to suggesting very high-cost outliers. This has two main causes: (a) The $\ell_1$ norm used to create the counterfactuals does not accurately represent real world effort. For example this norm considers any move in region to cost the same regardless of actual distance. (b) Because counterfactuals do not use a target set, they are prone to "overshooting" the desired goal. For example $CF_1$ resulted in a $95\%$ chance of passing the BAR when our goal was only $85\%$.

**Comparison of Amicable Perturbations vs. Other Approaches** We now compare amicable perturbations, counterfactuals Wachter et al. (2017); Mothilal et al. (2020) and CW attacks Carlini & Wagner (2017b) over the entire data sets. In Figure 5: Each bar chart refers to a particular data set and desired distance $\delta$ to the target set $T$. Each bar shows the percentage of individuals that a method was able to move inside the goal $\delta$ at a variety of costs $\epsilon$. (Bar charts for all data sets may be found in the Appendix.) The table summarizes this information for all data sets with the upper (red) value in each cell representing the data before the verification procedure and the lower (green) value the success rate after the verification procedure. Consider the bar chart on the top middle which refers to the German Credit data and a goal of $\delta = 0.5$ from the target (the same information as the last three columns of the table). At a $\epsilon = 0$ Deutsche Marks (DM) cost, amicable perturbations are able to move $73\%$ of individuals within the goal range by closing empty accounts. Counterfactuals do not match this success until the cost $\epsilon = 7,000$DM, and CW attacks never achieve more than a $31\%$ success rate. Amicable perturbations outperform counterfactuals in all of the test scenarios.

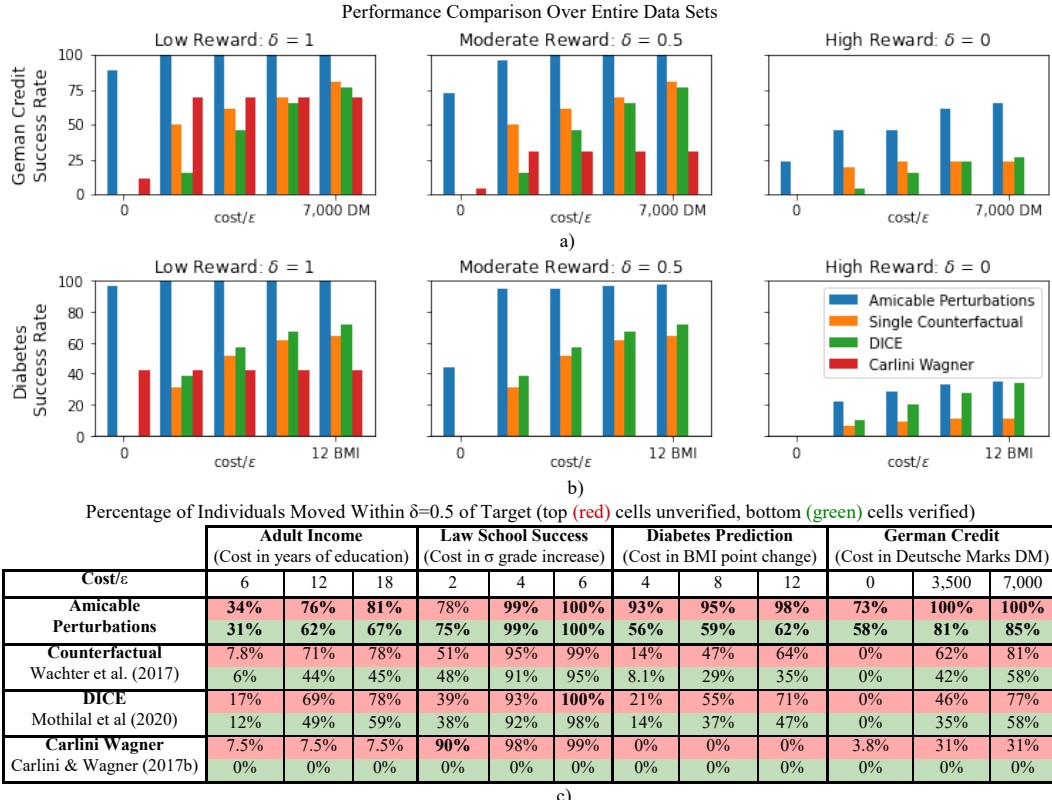

Figure 5: a) & b) show average success rate for moving individuals within a variety of distances ($\delta$) to the target set. The y-axis shows the percentage of individuals within the goal distance, and the x-axis, represents different costs ($\epsilon$ values). c) Summarizes success values for all data sets. The upper (red) value for each row is the success rate before the verification procedure and the lower (green) value is the success rate after verification with a 10% chance of rejecting valid examples.

**Impact and Effectiveness of Verifier:** The first important take away from the success rates after verification is that the verifier was 100% effective at eliminating Carlini Wagner adversarial examples (visible in the bottom row of the table in Figure 5 c), implying that the verification method does indeed eliminate inputs that fool the classifier. Importantly, the verification procedure also removes a significant number of amicable perturbations and counterfactuals. Consider the second column of Figure 5 c: Out of all amicable perturbations generated 14% appeared effective but where eliminated by the verification procedure. Counterfactual methods fared even worse with 20% to 27% of counterfactuals eliminated. This reinforces the necessity of a verification procedure.

**Concluding Remarks & Future Work** In this work, we proposed amicable perturbations which find efficient actions to an input in order to change an undesirable classification. We discussed the key distinctions from adversarial attacks and counterfactuals. Our proposed framework measures the cost of actions, their real world feasibility, and how close the actions come to achieving a goal in principled manner, allowing for truly optimal advice. We also developed a novel method for verifying that amicable perturbations affect the true class probabilities and don't just fool the classifier. Finally, we showed a comprehensive evaluation on data sets from multiple fields demonstrating their effectiveness. We finally note the universal applicability of amicable perturbations in generating "feasible advice" whenever the decision of an ML classifier is undesirable. Our framework is general enough for future work to incorporate additional desiderata. Of particular note is the concept of causality. $\mathcal{A}(\mathbf{x})$ could be modified to include casual relationships in inputs (e.g. education cannot be increased without increasing age) as suggested by Karimi et al. (2020). König et al. (2023) noted that a classifier could pick up on correlations that have no causal relationship with the label which could result in changes not having the anticipated effect. They suggested using Structural Causal Models (SCMs) to ensure causal relationships which could also be incorporated into our verification procedure. Ensuring causality through the use of longitudinal data on the actual effects of amicable perturbations could also be incorporated into our framework and verification procedure.

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

## 5 APPENDIX

### 5.1 ANALYSIS OF $d_{\mathcal{Y}}$

#### 5.1.1 PROOF OF THEOREM 1

Recall that our target sets have the form

$$T = \left\{ \mathbf{z} \in \mathcal{Y} \,\middle|\, \sum_{i \in \mathcal{W}} z_i \geq p \,,\, \sum_{i \in \mathcal{U}} z_i \leq q \right\},$$

where either $\mathcal{W}$ or $\mathcal{U}$ could be empty. Also recall

$$d_{\mathcal{Y}}(\tilde{\mathbf{y}}, T) = \min_{\mathbf{z} \in T} D_f(\tilde{\mathbf{y}} \| \mathbf{z})$$

$$= \min_{\mathbf{z} \in T} \sum_{i=1}^{k} z_i f \left( \frac{\tilde{y}_i}{z_i} \right). \tag{7}$$

*Closed Form:*

Our proof will be made easier by introducing notation $\mathcal{N} = (\mathcal{W} \cup \mathcal{U})^C$ as the neutral classes that are neither desirable nor undesirable. We will use the fact that $1 = \mathcal{S}_{\mathcal{W}} + \mathcal{S}_{\mathcal{U}} + \mathcal{S}_{\mathcal{N}}$ to rewrite equation 6 as

$$d_{\mathcal{Y}}(\tilde{\mathbf{y}}, T) = \begin{cases} 0 & \text{if } \mathcal{S}_{\mathcal{W}} \geq p \text{ and } \mathcal{S}_{\mathcal{U}} \leq q \\ pf\left(\frac{\mathcal{S}_{\mathcal{W}}}{p}\right) + (1-p)f\left(\frac{\mathcal{S}_{\mathcal{U}}+\mathcal{S}_{\mathcal{N}}}{1-p}\right) & \text{if } \mathcal{S}_{\mathcal{W}} < p \text{ and } \mathcal{S}_{\mathcal{U}} \leq (1-\mathcal{S}_{\mathcal{W}})\left(\frac{q}{1-p}\right) \\ qf\left(\frac{\mathcal{S}_{\mathcal{U}}}{q}\right) + (1-q)f\left(\frac{\mathcal{S}_{\mathcal{W}}+\mathcal{S}_{\mathcal{N}}}{1-q}\right) & \text{if } \mathcal{S}_{\mathcal{U}} > q \text{ and } \mathcal{S}_{\mathcal{W}} \geq (1-\mathcal{S}_{\mathcal{U}})\left(\frac{p}{1-q}\right) \\ pf\left(\frac{\mathcal{S}_{\mathcal{W}}}{p}\right) + qf\left(\frac{\mathcal{S}_{\mathcal{U}}}{q}\right) + (1-p-q)f\left(\frac{\mathcal{S}_{\mathcal{N}}}{1-p-q}\right) & \text{if } \mathcal{S}_{\mathcal{U}} > (1-\mathcal{S}_{\mathcal{W}})\left(\frac{q}{1-p}\right) \\ & \text{and } \mathcal{S}_{\mathcal{W}} < (1-\mathcal{S}_{\mathcal{U}})\left(\frac{p}{1-q}\right) \end{cases},$$

where $\mathcal{S}_{\mathcal{W}} = \sum_{i \in \mathcal{W}} \tilde{y}_i$, $\mathcal{S}_{\mathcal{U}} = \sum_{i \in \mathcal{U}} \tilde{y}_i$ and $\mathcal{S}_{\mathcal{N}} = \sum_{i \in \mathcal{N}} \tilde{y}_i$.

The case where $\tilde{\mathbf{y}} \in T$ is obvious so we consider only the case where $\tilde{\mathbf{y}} \notin T$, First note that $f$-divergence $D_f(\tilde{\mathbf{y}} \| \mathbf{z})$ is convex in $\mathbf{z}$. Furthermore $T$ is a convex set. Therefore any $\mathbf{z}$ satisfying the

KKT conditions is a minimizer. The KKT conditions for this problem can be written as

$$\nabla \mathcal{L}(\mathbf{z}) = \vec{0} \tag{8}$$

$$\sum_{i=1}^{k} z_i = 1 \tag{9}$$

$$p - \sum_{i \in \mathcal{W}} z_i \leq 0 \tag{10}$$

$$\sum_{i \in \mathcal{U}} z_i - q \leq 0 \tag{11}$$

$$\mu_1, \mu_2 \geq 0 \tag{12}$$

$$\mu_1 \left( p - \sum_{i \in \mathcal{W}} z_i \right) = 0 \tag{13}$$

$$\mu_2 \left( q - \sum_{i \in \mathcal{U}} z_i \right) = 0, \tag{14}$$

where Lagrangian is defined by

$$\mathcal{L}(\mathbf{z}) = \sum_{i=1}^{k} z_i f \left( \frac{\tilde{y}_i}{z_i} \right) + \lambda \sum_{i=1}^{k} z_i + \mu_1 \left( p - \sum_{i \in \mathcal{W}} z_i \right) + \mu_2 \left( \sum_{i \in \mathcal{U}} z_i - q \right).$$

Note that we have neglected to explicitly state the requirement that $0 \leq z_i \leq 1$ for all $i$. This is because our eventual solution will satisfy these bounds anyways, and omitting these bounds will drastically simplify our calculations. We now rewrite equation 8 as

$$f \left( \frac{\tilde{y}_i}{z_i} \right) - \frac{\tilde{y}_i}{z_i} f' \left( \frac{\tilde{y}_i}{z_i} \right) + \lambda - \mu_1 = 0 \qquad\qquad i \in \mathcal{W} \tag{15}$$

$$f \left( \frac{\tilde{y}_i}{z_i} \right) - \frac{\tilde{y}_i}{z_i} f' \left( \frac{\tilde{y}_i}{z_i} \right) + \lambda + \mu_2 = 0 \qquad\qquad i \in \mathcal{U} \tag{16}$$

$$f \left( \frac{\tilde{y}_i}{z_i} \right) - \frac{\tilde{y}_i}{z_i} f' \left( \frac{\tilde{y}_i}{z_i} \right) + \lambda = 0 \qquad\qquad i \in \mathcal{N} \tag{17}$$

We now propose a solution can be found where that the ratios $\frac{\tilde{y}_i}{z_i}$ are constant in each of the sets $\mathcal{W}$, $\mathcal{U}$, $\mathcal{N}$. That is

$$
\begin{aligned}
z_i &= C_{\mathcal{W}} \tilde{y}_i & i \in \mathcal{W} \\
z_i &= C_{\mathcal{U}} \tilde{y}_i & i \in \mathcal{U} \\
z_i &= C_{\mathcal{N}} \tilde{y}_i & i \in \mathcal{N}.
\end{aligned}
$$

In that case we can satisfy conditions equation 15, equation 16 and equation 17 (originally equation 8) by setting

$$
\begin{aligned}
\lambda &= C_{\mathcal{N}}^{-1} f'(C_{\mathcal{N}}^{-1}) - f(C_{\mathcal{N}}^{-1}) \\
\mu_1 &= \lambda + f(C_{\mathcal{W}}^{-1}) - C_{\mathcal{W}}^{-1} f'(C_{\mathcal{W}}^{-1}) \\
\mu_2 &= -\lambda - f(C_{\mathcal{U}}^{-1}) + C_{\mathcal{U}}^{-1} f'(C_{\mathcal{U}}^{-1}).
\end{aligned}
$$

We can now reformulate equation 12 so that it is easier to analyze. We will first define $h(x) = x f'(x) - f(x)$. Note that because $f(x)$ is convex $h'(x) = x f''(x) \geq 0$ for all $x \geq 0$ and $h(x)$ is increasing. We can then rewrite our formulas for $\lambda$, $\mu_1$ and $\mu_2$.

$$
\begin{aligned}
\lambda &= h(C_{\mathcal{N}}^{-1}) \\
\mu_1 &= h(C_{\mathcal{N}}^{-1}) - h(C_{\mathcal{W}}^{-1}) \\
\mu_2 &= h(C_{\mathcal{U}}^{-1}) - h(C_{\mathcal{N}}^{-1})
\end{aligned}
$$

Then $\mu_1 \geq 0$ becomes

$$h(C_\mathcal{N}^{-1}) \geq h(C_\mathcal{W}^{-1})$$
$$C_\mathcal{N}^{-1} \geq C_\mathcal{W}^{-1}$$
$$C_\mathcal{N} \leq C_\mathcal{W},$$

and $\mu_2 \geq 0$ similarly becomes $C_\mathcal{N} \geq C_\mathcal{U}$. This means equation 12 is equivalent to

$$C_\mathcal{U} \leq C_\mathcal{N} \leq C_\mathcal{W} \tag{18}$$

We must now find values of $C_\mathcal{W}$, $C_\mathcal{U}$ and $C_\mathcal{N}$ that satisfy equation 9 through equation 14. We will consider 3 cases illustrated in figure 5.1.1.

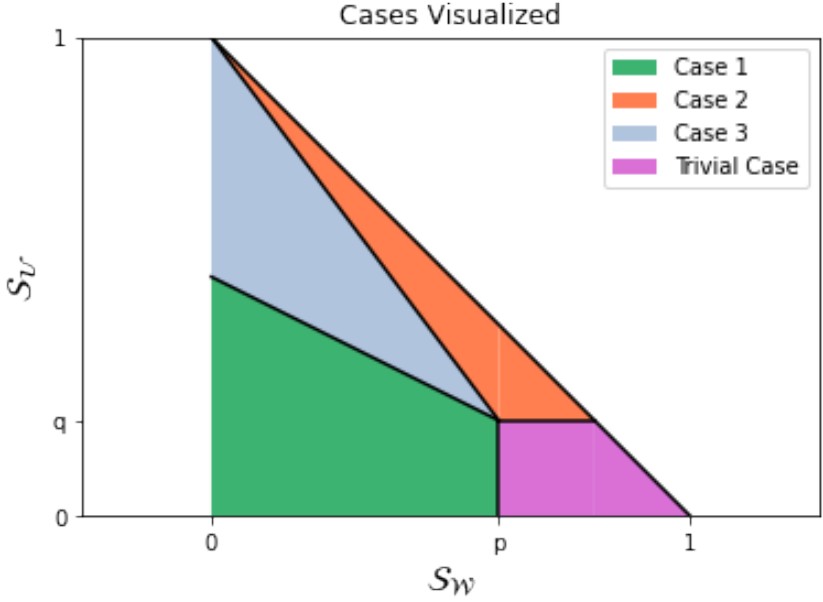

Figure 6: The three cases visualized in probability space.

**Case: 1** Suppose $\mathcal{S}_\mathcal{W} < p$ and $\mathcal{S}_\mathcal{U} \leq (1 - \mathcal{S}_\mathcal{W}) \left( \frac{q}{1-p} \right)$.

Let $C_\mathcal{W} = \frac{p}{\mathcal{S}_\mathcal{W}}$ and $C_\mathcal{U} = C_\mathcal{N} = \frac{1-p}{\mathcal{S}_\mathcal{U}+\mathcal{S}_\mathcal{N}}$. This implies $\mu_2 = 0$ which satisfies equation 14 and half of equation 12. This also implies $\sum_{i=\in\mathcal{W}} z_i = p$ satisfying equation 10 and equation 13. We will use the fact $\mathcal{S}_\mathcal{U} + \mathcal{S}_\mathcal{N} = 1 - \mathcal{S}_\mathcal{W}$ in our proof of condition equation 11.

$$\sum_{i\in\mathcal{S}_\mathcal{U}} z_i = \sum_{i\in\mathcal{S}_\mathcal{U}} C_\mathcal{U} \tilde{y}_i$$
$$= \frac{1-p}{\mathcal{S}_\mathcal{U} + \mathcal{S}_\mathcal{N}} \mathcal{S}_\mathcal{U}$$
$$\leq \frac{1-p}{\mathcal{S}_\mathcal{U} + \mathcal{S}_\mathcal{N}} (1 - \mathcal{S}_\mathcal{W}) \left( \frac{q}{1-p} \right)$$
$$= q$$

This proves equation 11 is satisfied.

Because $\mathcal{S}_\mathcal{W} < p$ we have

$$C_\mathcal{W} = \frac{p}{\mathcal{S}_\mathcal{W}} > 1 > \frac{1-p}{1-\mathcal{S}_\mathcal{W}} = \frac{1-p}{\mathcal{S}_\mathcal{U} + \mathcal{S}_\mathcal{N}} = C_\mathcal{N}$$

This implies $\mu_1 > 0$ and satisfies the other half of equation 12.

We have now shown all the KKT conditions are satisfied and we have found a minimizer. We now plug these values into equation 7 to find a closed form for the distance.

$$
\begin{aligned}
d_{\mathcal{Y}}(\tilde{\mathbf{y}}, T) &= \min_{\mathbf{z} \in T} \sum_{i=1}^{k} z_i f\left(\frac{\tilde{y}_i}{z_i}\right) \\
&= \sum_{i \in \mathcal{W}} \frac{p\tilde{y}_i}{\mathcal{S}_{\mathcal{W}}} f\left(\frac{\mathcal{S}_{\mathcal{W}}}{p}\right) + \sum_{i \notin \mathcal{W}} \frac{(1-p)\tilde{y}_i}{\mathcal{S}_{\mathcal{U}} + \mathcal{S}_{\mathcal{N}}} f\left(\frac{\mathcal{S}_{\mathcal{U}} + \mathcal{S}_{\mathcal{N}}}{1-p}\right) \\
&= pf\left(\frac{\mathcal{S}_{\mathcal{W}}}{p}\right) + (1-p)f\left(\frac{\mathcal{S}_{\mathcal{U}} + \mathcal{S}_{\mathcal{N}}}{1-p}\right).
\end{aligned}
$$

**Case: 2** Suppose $\mathcal{S}_{\mathcal{U}} > q$ and $\mathcal{S}_{\mathcal{W}} \geq (1 - \mathcal{S}_{\mathcal{U}})\left(\frac{p}{1-q}\right)$.

Let $C_{\mathcal{U}} = \frac{q}{\mathcal{S}_{\mathcal{U}}}$ and $C_{\mathcal{W}} = C_{\mathcal{N}} = \frac{1-q}{\mathcal{S}_{\mathcal{W}} + \mathcal{S}_{\mathcal{N}}}$. This implies $\mu_1 = 0$ which satisfies equation 13 and half of equation 12. We also have $\sum_{i=\in\mathcal{U}} z_i = q$ satisfying equation 11 and equation 14. We now prove condition equation 10 is satisfied.

$$
\begin{aligned}
\sum_{i \in \mathcal{S}_{\mathcal{W}}} z_i &= \sum_{i \in \mathcal{S}_{\mathcal{W}}} C_{\mathcal{W}} \tilde{y}_i \\
&= \frac{1-q}{\mathcal{S}_{\mathcal{W}} + \mathcal{S}_{\mathcal{N}}} \mathcal{S}_{\mathcal{W}} \\
&\geq \frac{1-q}{\mathcal{S}_{\mathcal{W}} + \mathcal{S}_{\mathcal{N}}} (1 - \mathcal{S}_{\mathcal{U}})\left(\frac{p}{1-q}\right) \\
&= p
\end{aligned}
$$

Finally we prove $C_{\mathcal{N}} \geq C_{\mathcal{U}}$ implying $\mu_2 \geq 0$ which satisfies the other half of equation 12

$$
C_{\mathcal{U}} = \frac{q}{\mathcal{S}_{\mathcal{U}}} < 1 < \frac{1-q}{1-\mathcal{S}_{\mathcal{U}}} = \frac{1-q}{\mathcal{S}_{\mathcal{W}} + \mathcal{S}_{\mathcal{N}}} = C_{\mathcal{N}}
$$

Now that we have proven that this is a minimizer we will again plug solution into equation 7 to find the distance value.

$$
\begin{aligned}
d_{\mathcal{Y}}(\tilde{\mathbf{y}}, T) &= \min_{\mathbf{z} \in T} \sum_{i=1}^{k} z_i f\left(\frac{\tilde{y}_i}{z_i}\right) \\
&= \sum_{i \in \mathcal{U}} \frac{q\tilde{y}_i}{\mathcal{S}_{\mathcal{U}}} f\left(\frac{\mathcal{S}_{\mathcal{U}}}{q}\right) + \sum_{i \notin \mathcal{U}} \frac{(1-q)\tilde{y}_i}{\mathcal{S}_{\mathcal{W}} + \mathcal{S}_{\mathcal{N}}} f\left(\frac{\mathcal{S}_{\mathcal{W}} + \mathcal{S}_{\mathcal{N}}}{1-p}\right) \\
&= qf\left(\frac{\mathcal{S}_{\mathcal{U}}}{q}\right) + (1-q)f\left(\frac{\mathcal{S}_{\mathcal{W}} + \mathcal{S}_{\mathcal{N}}}{1-q}\right).
\end{aligned}
$$

**Case: 3** Suppose $\mathcal{S}_{\mathcal{U}} > (1 - \mathcal{S}_{\mathcal{W}})\left(\frac{q}{1-p}\right)$ and $\mathcal{S}_{\mathcal{W}} < (1 - \mathcal{S}_{\mathcal{U}})\left(\frac{p}{1-q}\right)$.

Let $C_{\mathcal{W}} = \frac{p}{\mathcal{S}_{\mathcal{W}}}$, $C_{\mathcal{U}} = \frac{q}{\mathcal{S}_{\mathcal{U}}}$ and $C_{\mathcal{N}} = \frac{1-p-q}{\mathcal{S}_{\mathcal{N}}}$ in which case $\sum_{i=\in\mathcal{W}} z_i = p$ (satisfying equation 10 and equation 13), $\sum_{i=\in\mathcal{U}} z_i = q$ (satisfying equation 11 and equation 14). The choice of $C_{\mathcal{N}}$

ensures that equation 9 is satisfied:

$$\sum_{i=1}^{M} z_i = \sum_{i \in C_{\mathcal{W}}} z_i + \sum_{i \in C_{\mathcal{U}}} z_i + \sum_{i \in C_{\mathcal{N}}} z_i$$
$$= \sum_{i \in C_{\mathcal{W}}} C_{\mathcal{W}} \tilde{y}_i + \sum_{i \in C_{\mathcal{U}}} C_{\mathcal{U}} \tilde{y}_i + \sum_{i \in C_{\mathcal{N}}} C_{\mathcal{N}} \tilde{y}_i$$
$$= C_{\mathcal{W}} \mathcal{S}_{\mathcal{W}} + C_{\mathcal{U}} \mathcal{S}_{\mathcal{U}} + C_{\mathcal{N}} \mathcal{S}_{\mathcal{N}}$$
$$= 1$$

Finally we show that equation 12 is satisfied. Consider

$$\mathcal{S}_{\mathcal{U}} > (1 - \mathcal{S}_{\mathcal{W}}) \left( \frac{q}{1-p} \right)$$
$$\mathcal{S}_{\mathcal{U}}(1-p) > (1 - \mathcal{S}_{\mathcal{W}})q$$
$$\mathcal{S}_{\mathcal{U}} - p\mathcal{S}_{\mathcal{U}} - q\mathcal{S}_{\mathcal{U}} > q - q\mathcal{S}_{\mathcal{W}} - q\mathcal{S}_{\mathcal{U}}$$
$$\mathcal{S}_{\mathcal{U}}(1 - p - q) > q\mathcal{S}_{\mathcal{N}}$$
$$\frac{1-p-q}{\mathcal{S}_{\mathcal{N}}} > \frac{q}{\mathcal{S}_{\mathcal{U}}}$$
$$C_{\mathcal{N}} > C_{\mathcal{U}}$$

and

$$\mathcal{S}_{\mathcal{W}} < (1 - \mathcal{S}_{\mathcal{U}}) \left( \frac{p}{1-q} \right)$$
$$\mathcal{S}_{\mathcal{W}}(1-q) < (1 - \mathcal{S}_{\mathcal{U}})p$$
$$\mathcal{S}_{\mathcal{W}} - q\mathcal{S}_{\mathcal{W}} - p\mathcal{S}_{\mathcal{U}} < p - p\mathcal{S}_{\mathcal{W}} - p\mathcal{S}_{\mathcal{U}}$$
$$\mathcal{S}_{\mathcal{W}}(1 - p - q) < p\mathcal{S}_{\mathcal{N}}$$
$$\frac{1-p-q}{\mathcal{S}_{\mathcal{N}}} < \frac{p}{\mathcal{S}_{\mathcal{W}}}$$
$$C_{\mathcal{N}} < C_{\mathcal{W}}.$$

This proves equation 18 which is equivalent to equation 12 Plugging these minimizing values of $\mathbf{z}$ into equation 7 yields

$$d_{\mathcal{Y}}(\tilde{\mathbf{y}}, T) = \min_{\mathbf{z} \in T} \sum_{i=1}^{k} z_i f \left( \frac{\tilde{y}_i}{z_i} \right)$$
$$= \sum_{i \in \mathcal{W}} \frac{p\tilde{y}_i}{\mathcal{S}_{\mathcal{W}}} f \left( \frac{\mathcal{S}_{\mathcal{W}}}{p} \right) + \sum_{i \in \mathcal{U}} \frac{q\tilde{y}_i}{\mathcal{S}_{\mathcal{U}}} f \left( \frac{\mathcal{S}_{\mathcal{U}}}{q} \right) + \sum_{i \in \mathcal{W}} \frac{(1-p-q)\tilde{y}_i}{\mathcal{S}_{\mathcal{N}}} f \left( \frac{\mathcal{S}_{\mathcal{N}}}{1-p-q} \right)$$
$$= p f \left( \frac{\mathcal{S}_{\mathcal{W}}}{p} \right) + q f \left( \frac{\mathcal{S}_{\mathcal{U}}}{q} \right) + (1-p-q) f \left( \frac{\mathcal{S}_{\mathcal{N}}}{1-p-q} \right).$$

This proves our closed form solution.

We now show that this function is both continuous and continuously differentiable.

*Continuity:*

To prove continuity we need only show continuity the piece-wise boundaries which we will evaluate one at a time.

**Boundary 1:** $\mathcal{S}_{\mathcal{W}} = p$. The two functions that share this boundary are 0 and $p f \left( \frac{\mathcal{S}_{\mathcal{W}}}{p} \right) + (1 - p) f \left( \frac{1 - \mathcal{S}_{\mathcal{W}}}{1-p} \right)$. Plugging the boundary into the latter function yields

$$p f \left( \frac{\mathcal{S}_{\mathcal{W}}}{p} \right) + (1-p) f \left( \frac{1 - \mathcal{S}_{\mathcal{W}}}{1-p} \right) = p f \left( \frac{p}{p} \right) + (1-p) f \left( \frac{1-p}{1-p} \right)$$
$$= 0.$$

The two functions are equal on the boundary and the boundary is continuous.

**Boundary 2:** $\mathcal{S}_\mathcal{U} = q$. The two functions that share this boundary are $0$ and $qf\left(\frac{\mathcal{S}_\mathcal{U}}{q}\right) + (1 - q)f\left(\frac{1-\mathcal{S}_\mathcal{U}}{1-q}\right)$. Plugging the boundary into the latter function yields

$$qf\left(\frac{\mathcal{S}_\mathcal{U}}{q}\right) + (1-q)f\left(\frac{1-\mathcal{S}_\mathcal{U}}{1-q}\right) = qf\left(\frac{q}{q}\right) + (1-q)f\left(\frac{1-q}{1-q}\right)$$
$$= 0.$$

The two functions are equal on the boundary and the boundary is continuous.

**Boundary 3:** $\mathcal{S}_\mathcal{U} = (1 - \mathcal{S}_\mathcal{W})\left(\frac{q}{1-p}\right)$. The two functions that share this boundary are $pf\left(\frac{\mathcal{S}_\mathcal{W}}{p}\right) + (1-p)f\left(\frac{1-\mathcal{S}_\mathcal{W}}{1-p}\right)$ and $pf\left(\frac{\mathcal{S}_\mathcal{W}}{p}\right) + qf\left(\frac{\mathcal{S}_\mathcal{U}}{q}\right) + (1-p-q)f\left(\frac{\mathcal{S}_\mathcal{N}}{1-p-q}\right)$. Plugging the boundary into the latter function yields

$$pf\left(\frac{\mathcal{S}_\mathcal{W}}{p}\right) + qf\left(\frac{\mathcal{S}_\mathcal{U}}{q}\right) + (1-p-q)f\left(\frac{1-\mathcal{S}_\mathcal{W}-\mathcal{S}_\mathcal{U}}{1-p-q}\right) = pf\left(\frac{\mathcal{S}_\mathcal{W}}{p}\right) + (1-p)f\left(\frac{1-\mathcal{S}_\mathcal{W}}{1-p}\right).$$

The two functions are equal on the boundary and the boundary is continuous.

**Boundary 4:** $\mathcal{S}_\mathcal{W} = (1 - \mathcal{S}_\mathcal{U})\left(\frac{p}{1-q}\right)$. The two functions that share this boundary are $qf\left(\frac{\mathcal{S}_\mathcal{U}}{q}\right) + (1-q)f\left(\frac{1-\mathcal{S}_\mathcal{U}}{1-q}\right)$ and $pf\left(\frac{\mathcal{S}_\mathcal{W}}{p}\right) + qf\left(\frac{\mathcal{S}_\mathcal{U}}{q}\right) + (1-p-q)f\left(\frac{\mathcal{S}_\mathcal{N}}{1-p-q}\right)$. Plugging the boundary into the latter function yields

$$qf\left(\frac{\mathcal{S}_\mathcal{U}}{q}\right) + pf\left(\frac{\mathcal{S}_\mathcal{W}}{p}\right) + (1-p-q)f\left(\frac{1-\mathcal{S}_\mathcal{U}-\mathcal{S}_\mathcal{W}}{1-p-q}\right) = qf\left(\frac{\mathcal{S}_\mathcal{U}}{q}\right) + (1-q)f\left(\frac{1-\mathcal{S}_\mathcal{U}}{1-q}\right).$$

The two functions are equal on the boundary and the boundary is continuous.

We have now shown continuity on all boundaries and the function is continuous.

*Differentiability:*

Finally we show are function is continuously differentiable by showing all partial derivatives exist and are continuous. We use the closed form equation equation 6 found in the body of the paper (which is equivalent to the one found in the beginning of the proof) that suppresses $\mathcal{S}_\mathcal{N}$. This makes it easier to differentiate with respect to $\tilde{y}_i$, $i \in \mathcal{W} \cup \mathcal{U}$.

$$d_\mathcal{Y}(\tilde{\mathbf{y}}, T) = \begin{cases} 0 & \text{if } \mathcal{S}_\mathcal{W} \geq p \text{ and } \mathcal{S}_\mathcal{U} \leq q \\ pf\left(\frac{\mathcal{S}_\mathcal{W}}{p}\right) + (1-p)f\left(\frac{1-\mathcal{S}_\mathcal{W}}{1-p}\right) & \text{if } \mathcal{S}_\mathcal{W} < p \text{ and } \mathcal{S}_\mathcal{U} \leq (1-\mathcal{S}_\mathcal{W})\left(\frac{q}{1-p}\right) \\ qf\left(\frac{\mathcal{S}_\mathcal{U}}{q}\right) + (1-q)f\left(\frac{1-\mathcal{S}_\mathcal{U}}{1-q}\right) & \text{if } \mathcal{S}_\mathcal{U} > q \text{ and } \mathcal{S}_\mathcal{W} \geq (1-\mathcal{S}_\mathcal{U})\left(\frac{p}{1-q}\right) \\ pf\left(\frac{\mathcal{S}_\mathcal{W}}{p}\right) + qf\left(\frac{\mathcal{S}_\mathcal{U}}{q}\right) + (1-p-q)f\left(\frac{1-\mathcal{S}_\mathcal{W}-\mathcal{S}_\mathcal{U}}{1-p-q}\right) & \text{if } \mathcal{S}_\mathcal{U} > (1-\mathcal{S}_\mathcal{W})\left(\frac{q}{1-p}\right) \\ & \text{and } \mathcal{S}_\mathcal{W} < (1-\mathcal{S}_\mathcal{U})\left(\frac{p}{1-q}\right) \end{cases}$$

We now take the derivative with respect to a desirable class ($i \in \mathcal{W}$).

$$\frac{\partial}{\partial \tilde{y}_{i \in \mathcal{W}}} d_\mathcal{Y}(\tilde{\mathbf{y}}, T) = \begin{cases} 0 & \text{if } \mathcal{S}_\mathcal{W} > p \text{ and } \mathcal{S}_\mathcal{U} < q \\ f'\left(\frac{\mathcal{S}_\mathcal{W}}{p}\right) - f'\left(\frac{1-\mathcal{S}_\mathcal{W}}{1-p}\right) & \text{if } \mathcal{S}_\mathcal{W} < p \text{ and } \mathcal{S}_\mathcal{U} < (1-\mathcal{S}_\mathcal{W})\left(\frac{q}{1-p}\right) \\ 0 & \text{if } \mathcal{S}_\mathcal{U} > q \text{ and } \mathcal{S}_\mathcal{W} > (1-\mathcal{S}_\mathcal{U})\left(\frac{p}{1-q}\right) \\ f'\left(\frac{\mathcal{S}_\mathcal{W}}{p}\right) - f'\left(\frac{1-\mathcal{S}_\mathcal{W}-\mathcal{S}_\mathcal{U}}{1-p-q}\right) & \text{if } \mathcal{S}_\mathcal{U} > (1-\mathcal{S}_\mathcal{W})\left(\frac{q}{1-p}\right) \\ & \text{and } \mathcal{S}_\mathcal{W} < (1-\mathcal{S}_\mathcal{U})\left(\frac{p}{1-q}\right) \end{cases}$$

Now we need only ensure all pieces agree on the boundaries to shoe that the derivative exists and is continuous.

**Boundary 1:** $\mathcal{S}_\mathcal{W} = p$. The two functions that share this boundary are $0$ and $f'\left(\frac{\mathcal{S}_\mathcal{W}}{p}\right) - f'\left(\frac{1-\mathcal{S}_\mathcal{W}}{1-p}\right)$. Plugging the boundary into the latter function yields

$$
f'\left(\frac{\mathcal{S}_\mathcal{W}}{p}\right) - f'\left(\frac{1-\mathcal{S}_\mathcal{W}}{1-p}\right) = f'\left(\frac{p}{p}\right) - f'\left(\frac{1-p}{1-p}\right)
$$
$$
= f'(1) + f(1')
$$
$$
= 0.
$$

Then setting the derivative at the boundary to $0$ makes the derivative on this boundary continuous.

**Boundary 2:** $\mathcal{S}_\mathcal{U} = q$. The two functions that share this boundary are both $0$, and setting the derivative at the boundary to $0$ makes the derivative on this boundary continuous.

**Boundary 3:** $\mathcal{S}_\mathcal{U} = (1 - \mathcal{S}_\mathcal{W})\left(\frac{q}{1-p}\right)$. The two functions that share this boundary are $f'\left(\frac{\mathcal{S}_\mathcal{W}}{p}\right) - f'\left(\frac{1-\mathcal{S}_\mathcal{W}}{1-p}\right)$ and $f'\left(\frac{\mathcal{S}_\mathcal{W}}{p}\right) - f'\left(\frac{1-\mathcal{S}_\mathcal{W}-\mathcal{S}_\mathcal{U}}{1-p-q}\right)$. Plugging the boundary into the latter function yields

$$
f'\left(\frac{\mathcal{S}_\mathcal{W}}{p}\right) - f'\left(\frac{1-\mathcal{S}_\mathcal{W}-\mathcal{S}_\mathcal{U}}{1-p-q}\right) = f'\left(\frac{\mathcal{S}_\mathcal{W}}{p}\right) - f'\left(\frac{1-\mathcal{S}_\mathcal{W}}{1-p}\right)
$$

Then setting the derivative at the boundary to $f'\left(\frac{\mathcal{S}_\mathcal{W}}{p}\right) - f'\left(\frac{1-\mathcal{S}_\mathcal{W}}{1-p}\right)$ makes the derivative on this boundary continuous.

**Boundary 4:** $\mathcal{S}_\mathcal{W} = (1 - \mathcal{S}_\mathcal{U})\left(\frac{p}{1-q}\right)$. The two functions that share this boundary are $0$ and $f'\left(\frac{\mathcal{S}_\mathcal{W}}{p}\right) - f'\left(\frac{1-\mathcal{S}_\mathcal{W}-\mathcal{S}_\mathcal{U}}{1-p-q}\right)$. We rewrite the boundary as $\mathcal{S}_\mathcal{U} = \frac{1-q}{p}\mathcal{S}_\mathcal{W} + 1$ and plug it into the latter function.

$$
f'\left(\frac{\mathcal{S}_\mathcal{W}}{p}\right) - f'\left(\frac{1-\mathcal{S}_\mathcal{W}-\mathcal{S}_\mathcal{U}}{1-p-q}\right) = f'\left(\frac{\mathcal{S}_\mathcal{W}}{p}\right) - f'\left(\frac{1-\mathcal{S}_\mathcal{W} - \left(\frac{1-q}{p}\mathcal{S}_\mathcal{W}+1\right)}{1-p-q}\right)
$$
$$
= 0
$$

Then setting the derivative at the boundary to $0$ makes the derivative on this boundary continuous.

This yields the continuous partial derivative

$$
\frac{\partial}{\partial \tilde{y}_{i\in\mathcal{W}}} d_\mathcal{Y}(\tilde{\mathbf{y}}, T) = \begin{cases} 0 & \text{if } \mathcal{S}_\mathcal{W} \geq p \text{ and } \mathcal{S}_\mathcal{U} \leq q \\ f'\left(\frac{\mathcal{S}_\mathcal{W}}{p}\right) - f'\left(\frac{1-\mathcal{S}_\mathcal{W}}{1-p}\right) & \text{if } \mathcal{S}_\mathcal{W} < p \text{ and } \mathcal{S}_\mathcal{U} \leq (1-\mathcal{S}_\mathcal{W})\left(\frac{q}{1-p}\right) \\ 0 & \text{if } \mathcal{S}_\mathcal{U} > q \text{ and } \mathcal{S}_\mathcal{W} \geq (1-\mathcal{S}_\mathcal{U})\left(\frac{p}{1-q}\right) \\ f'\left(\frac{\mathcal{S}_\mathcal{W}}{p}\right) - f'\left(\frac{1-\mathcal{S}_\mathcal{W}-\mathcal{S}_\mathcal{U}}{1-p-q}\right) & \text{if } \mathcal{S}_\mathcal{U} > (1-\mathcal{S}_\mathcal{W})\left(\frac{q}{1-p}\right) \\ & \text{and } \mathcal{S}_\mathcal{W} < (1-\mathcal{S}_\mathcal{U})\left(\frac{p}{1-q}\right) \end{cases} .
$$

$$(19)$$

We now take the derivative with respect to a undesirable class ($i \in \mathcal{U}$).

$$
\frac{\partial}{\partial \tilde{y}_{i\in\mathcal{U}}} d_\mathcal{Y}(\tilde{\mathbf{y}}, T) = \begin{cases} 0 & \text{if } \mathcal{S}_\mathcal{W} > p \text{ and } \mathcal{S}_\mathcal{U} < q \\ 0 & \text{if } \mathcal{S}_\mathcal{W} < p \text{ and } \mathcal{S}_\mathcal{U} < (1-\mathcal{S}_\mathcal{W})\left(\frac{q}{1-p}\right) \\ f'\left(\frac{\mathcal{S}_\mathcal{U}}{q}\right) - f'\left(\frac{1-\mathcal{S}_\mathcal{U}}{1-q}\right) & \text{if } \mathcal{S}_\mathcal{U} > q \text{ and } \mathcal{S}_\mathcal{W} > (1-\mathcal{S}_\mathcal{U})\left(\frac{p}{1-q}\right) \\ f'\left(\frac{\mathcal{S}_\mathcal{U}}{q}\right) - f'\left(\frac{1-\mathcal{S}_\mathcal{W}-\mathcal{S}_\mathcal{U}}{1-p-q}\right) & \text{if } \mathcal{S}_\mathcal{U} > (1-\mathcal{S}_\mathcal{W})\left(\frac{q}{1-p}\right) \\ & \text{and } \mathcal{S}_\mathcal{W} < (1-\mathcal{S}_\mathcal{U})\left(\frac{p}{1-q}\right) \end{cases}
$$

Now we need only ensure that there is agreement on the boundaries.

**Boundary 1:** $\mathcal{S}_\mathcal{W} = p$. The two functions that share this boundary are both $0$, and setting the derivative at the boundary to $0$ makes the derivative on this boundary continuous.

**Boundary 2:** $\mathcal{S}_\mathcal{U} = q$. The two functions that share this boundary are both $0$ and $f'\left(\frac{\mathcal{S}_\mathcal{U}}{q}\right) - f'\left(\frac{1-\mathcal{S}_\mathcal{U}}{1-q}\right)$. Plugging the boundary into the latter function yields

$$f'\left(\frac{\mathcal{S}_\mathcal{U}}{q}\right) - f'\left(\frac{1-\mathcal{S}_\mathcal{U}}{1-q}\right) = f'\left(\frac{q}{q}\right) - f'\left(\frac{1-q}{1-q}\right)$$
$$= 0.$$

Then setting the derivative at the boundary to $0$ makes the derivative on this boundary continuous.

**Boundary 3:** $\mathcal{S}_\mathcal{U} = (1 - \mathcal{S}_\mathcal{W})\left(\frac{q}{1-p}\right)$. The two functions that share this boundary are $0$ and $f'\left(\frac{\mathcal{S}_\mathcal{U}}{q}\right) - f'\left(\frac{1-\mathcal{S}_\mathcal{W}-\mathcal{S}_\mathcal{U}}{1-p-q}\right)$. We rewrite the boundary as $\mathcal{S}_\mathcal{W} = 1 - \frac{1-p}{q}\mathcal{S}_\mathcal{U}$ and plug it into the latter function.

$$f'\left(\frac{\mathcal{S}_\mathcal{U}}{q}\right) - f'\left(\frac{1-\mathcal{S}_\mathcal{W}-\mathcal{S}_\mathcal{U}}{1-p-q}\right) = f'\left(\frac{\mathcal{S}_\mathcal{U}}{q}\right) - f'\left(\frac{1-\mathcal{S}_\mathcal{U}-\left(1-\frac{1-p}{q}\mathcal{S}_\mathcal{U}\right)}{1-p-q}\right)$$
$$= 0$$

Then setting the derivative at the boundary to $0$ makes the derivative on this boundary continuous.

**Boundary 4:** $\mathcal{S}_\mathcal{W} = (1 - \mathcal{S}_\mathcal{U})\left(\frac{p}{1-q}\right)$. The two functions that share this boundary are $f'\left(\frac{\mathcal{S}_\mathcal{U}}{q}\right) - f'\left(\frac{1-\mathcal{S}_\mathcal{U}}{1-q}\right)$ and $f'\left(\frac{\mathcal{S}_\mathcal{U}}{q}\right) - f'\left(\frac{1-\mathcal{S}_\mathcal{W}-\mathcal{S}_\mathcal{U}}{1-p-q}\right)$. Plugging the boundary into the latter function yields

$$f'\left(\frac{\mathcal{S}_\mathcal{U}}{q}\right) - f'\left(\frac{1-\mathcal{S}_\mathcal{W}-\mathcal{S}_\mathcal{U}}{1-p-q}\right) = f'\left(\frac{\mathcal{S}_\mathcal{U}}{q}\right) - f'\left(\frac{1-\mathcal{S}_\mathcal{U}-(1-\mathcal{S}_\mathcal{U})\left(\frac{p}{1-q}\right)}{1-p-q}\right)$$
$$= f'\left(\frac{\mathcal{S}_\mathcal{U}}{q}\right) - f'\left(\frac{1-\mathcal{S}_\mathcal{U}}{1-q}\right)$$

Then setting the derivative at the boundary to $f'\left(\frac{\mathcal{S}_\mathcal{U}}{q}\right) - f'\left(\frac{1-\mathcal{S}_\mathcal{U}}{1-q}\right)$ makes the derivative on this boundary continuous.

This yields the continuous partial derivative

$$\frac{\partial}{\partial \tilde{y}_{i\in\mathcal{U}}} d_\mathcal{Y}(\tilde{\mathbf{y}}, T) = \begin{cases} 0 & \text{if } \mathcal{S}_\mathcal{W} \geq p \text{ and } \mathcal{S}_\mathcal{U} \leq q \\ 0 & \text{if } \mathcal{S}_\mathcal{W} < p \text{ and } \mathcal{S}_\mathcal{U} \leq (1-\mathcal{S}_\mathcal{W})\left(\frac{q}{1-p}\right) \\ f'\left(\frac{\mathcal{S}_\mathcal{U}}{q}\right) - f'\left(\frac{1-\mathcal{S}_\mathcal{U}}{1-q}\right) & \text{if } \mathcal{S}_\mathcal{U} > q \text{ and } \mathcal{S}_\mathcal{W} \geq (1-\mathcal{S}_\mathcal{U})\left(\frac{p}{1-q}\right) \\ f'\left(\frac{\mathcal{S}_\mathcal{U}}{q}\right) - f'\left(\frac{1-\mathcal{S}_\mathcal{W}-\mathcal{S}_\mathcal{U}}{1-p-q}\right) & \text{if } \mathcal{S}_\mathcal{U} > (1-\mathcal{S}_\mathcal{W})\left(\frac{q}{1-p}\right) \\ & \text{and } \mathcal{S}_\mathcal{W} < (1-\mathcal{S}_\mathcal{U})\left(\frac{p}{1-q}\right) \end{cases}.$$

$$(20)$$

### 5.1.2 ADDITIONAL RESULTS

The following lemma shows that $d_\mathcal{Y}$ exhibits desirable behavior for any $f$-divergence if we restrict ourselves to the binary classification setting.

**Lemma 1** *In the binary classification environment, if $T = \{\mathbf{z} \in \mathcal{Y}|z_1 \geq p\}$, then $d_\mathcal{Y}(\tilde{\mathbf{y}}, T)$ is decreasing (not necessarily strictly) in $\tilde{\mathbf{y}}_1$ for $D(\tilde{\mathbf{y}}||\mathbf{z})$ any $f$-divergence.*

We prove Lemma 1.

Recall $d_{\mathcal{Y}}(\tilde{\mathbf{y}}, T) = \min_{\mathbf{z} \in T} D_f(\tilde{\mathbf{y}} || \mathbf{z})$. For binary probability distributions $\mathbf{a}$ and $\mathbf{b}$, the $f$-divergence has the simple form

$$D_f(\mathbf{b} || \mathbf{a}) = \mathbf{a}_1 f\left(\frac{\mathbf{b}_1}{\mathbf{a}_1}\right) + (1 - \mathbf{a}_1) f\left(\frac{1 - \mathbf{b}_1}{1 - \mathbf{a}_1}\right) \tag{21}$$

for a convex function $f$ with $f(1) = 0$. We show a relationship between this formula and a secant line. To refer to the secant line of a function $g(x)$ from point $x = \alpha$ to $x = \beta$ evaluated at $\gamma$, we will use the notation $S_g(\alpha, \beta; \gamma)$. When using this notation we will assume that $\alpha \leq \beta$.

We assume $\mathbf{a}_1 > \mathbf{b}_1$ and show that $D_f(\mathbf{b} || \mathbf{a})$ is equivalent to the secant line of $f(x)$ from $x = \frac{\mathbf{b}_1}{\mathbf{a}_1}$ to $\frac{1 - \mathbf{b}_1}{1 - \mathbf{a}_1}$ evaluated at 1. (Note $\frac{\mathbf{b}_1}{\mathbf{a}_1} < 1 < \frac{1 - \mathbf{b}_1}{1 - \mathbf{a}_1}$.) We show this simply using the point slope form.

$$S_f\left(\frac{\mathbf{b}_1}{\mathbf{a}_1}, \frac{1 - \mathbf{b}_1}{1 - \mathbf{a}_1}; x\right) = \left(x - \frac{1 - \mathbf{b}_1}{1 - \mathbf{a}_1}\right) \frac{f\left(\frac{1 - \mathbf{b}_1}{1 - \mathbf{a}_1}\right) - f\left(\frac{\mathbf{b}_1}{\mathbf{a}_1}\right)}{\frac{1 - \mathbf{b}_1}{1 - \mathbf{a}_1} - \frac{\mathbf{b}_1}{\mathbf{a}_1}} + f\left(\frac{1 - \mathbf{b}_1}{1 - \mathbf{a}_1}\right)$$

$$S_f\left(\frac{\mathbf{b}_1}{\mathbf{a}_1}, \frac{1 - \mathbf{b}_1}{1 - \mathbf{a}_1}; 1\right) = \left(1 - \frac{1 - \mathbf{b}_1}{1 - \mathbf{a}_1}\right) \frac{f\left(\frac{1 - \mathbf{b}_1}{1 - \mathbf{a}_1}\right) - f\left(\frac{\mathbf{b}_1}{\mathbf{a}_1}\right)}{\frac{1 - \mathbf{b}_1}{1 - \mathbf{a}_1} - \frac{\mathbf{b}_1}{\mathbf{a}_1}} + f\left(\frac{1 - \mathbf{b}_1}{1 - \mathbf{a}_1}\right)$$

$$= \mathbf{a}_1 f\left(\frac{\mathbf{b}_1}{\mathbf{a}_1}\right) + (1 - \mathbf{a}_1) f\left(\frac{1 - \mathbf{b}_1}{1 - \mathbf{a}_1}\right)$$

$$= D_f(\mathbf{b} || \mathbf{a})$$

Now that $D_f(\mathbf{b} || \mathbf{a})$ is related to a secant line we prove a few facts about secant lines of convex functions. If $g$ is convex, then $S_g(\alpha, \beta; \gamma)$ is decreasing in $\alpha$ and increasing in $\beta$ whenever $\alpha < \gamma < \beta$. Recall that if $g$ is convex, then by definition for any $v_1 < v_2 < v_3$, we have

$$\frac{g(v_2) - g(v_1)}{v_2 - v_1} \leq \frac{g(v_3) - g(v_1)}{v_3 - v_1} \leq \frac{g(v_3) - g(v_2)}{v_3 - v_2}. \tag{22}$$

Then for any $\beta < \tilde{\beta}$ we have

$$S_g(\alpha, \beta; \gamma) = (\gamma - \alpha)m + g(\alpha) \tag{23}$$

$$S_g(\alpha, \tilde{\beta}; \gamma) = (\gamma - \alpha)\tilde{m} + g(\alpha) \tag{24}$$

for $\tilde{m} \geq m$. It follows that for any $\gamma \geq \alpha$

$$S_g(\alpha, \beta; x) \leq S_g(\alpha, \tilde{\beta}; x), \tag{25}$$

and $S_g(\alpha, \beta; x)$ is increasing in $\beta$.

A similar argument shows that $S_g(\alpha, \beta; x)$ is decreasing in $\alpha$ when $\gamma \leq \beta$.

We will use these facts to analyze $d_{\mathcal{Y}}(\tilde{\mathbf{y}}, T) = \min_{\mathbf{z} \in T} D_f(\tilde{\mathbf{y}} || \mathbf{z})$. The $f$-divergence between identical distributions is zero, so we have $d_{\mathcal{Y}}(\tilde{\mathbf{y}}, T) = 0$ whenever $\tilde{\mathbf{y}}_1 \geq p$. When $\tilde{\mathbf{y}}_1 < p$ we have $\frac{\tilde{\mathbf{y}}_1}{\mathbf{z}_1} < 1 < \frac{1 - \tilde{\mathbf{y}}_1}{1 - \mathbf{z}_1}$ and

$$d_{\mathcal{Y}}(\tilde{\mathbf{y}}, T) = \min_{\mathbf{z} \in T} D_f(\tilde{\mathbf{y}} || \mathbf{z})$$

$$= \min_{\mathbf{z} \in T} S_f\left(\frac{\tilde{\mathbf{y}}_1}{\mathbf{z}_1}, \frac{1 - \tilde{\mathbf{y}}_1}{1 - \mathbf{z}_1}; 1\right),$$

which is decreasing in $\frac{\tilde{\mathbf{y}}_1}{\mathbf{z}_1}$ and increasing in $\frac{1 - \tilde{\mathbf{y}}_1}{1 - \mathbf{z}_1}$, so to achieve the minimum we use the smallest possible $\mathbf{z}_1$, i.e. $\mathbf{z}_1 = p$. We may now simplify

$$d_{\mathcal{Y}}(\tilde{\mathbf{y}}, T) = \begin{cases} S_f\left(\frac{\tilde{\mathbf{y}}_1}{p}, \frac{1 - \tilde{\mathbf{y}}_1}{1 - p}; 1\right) & \text{if } \tilde{\mathbf{y}} < p \\ 0 & \text{if } \tilde{\mathbf{y}} \geq p \end{cases}.$$

Note that this is continuous at $\tilde{\mathbf{y}} = p$ because $S_f(1, 1; 1) = f(1) = 0$. With this closed form solution for $d_{\mathcal{Y}}(\tilde{\mathbf{y}}, T)$ we may finish the proof.

We have already shown that $S_f\left(\frac{\tilde{\mathbf{y}}_1}{p}, \frac{1-\tilde{\mathbf{y}}_1}{1-p}; 1\right)$ is decreasing in $\frac{\tilde{\mathbf{y}}_1}{p}$ and increasing in $\frac{1-\tilde{\mathbf{y}}_1}{1-p}$, so increasing $\tilde{\mathbf{y}}_1$ will decrease $S_f\left(\frac{\tilde{\mathbf{y}}_1}{p}, \frac{1-\tilde{\mathbf{y}}_1}{1-p}; 1\right)$ and $d_{\mathcal{Y}}(\tilde{\mathbf{y}}, T)$ is decreasing in $\tilde{\mathbf{y}}_1$.

We now present a corollary to Theorem 1 that shows explicitly that $d_{\mathcal{Y}}$ decreases with added probability to the desirable classes and increases with added probability to the undesirable classes.

**Corollary 1** *If $T$ is of form 1 and $f$ is twice differentiable, then $d_{\mathcal{Y}}(\tilde{\mathbf{y}}, T)$ is decreasing in $\tilde{\mathbf{y}}_i$ if $i \in \mathcal{W}$ and is increasing if $i \in \mathcal{U}$.*

To prove Corollary 1, we need only show equation equation 6 is decreasing in $\tilde{y}_i$ for $i \in \mathcal{W}$ and increasing in $\tilde{y}_i$ for $i \in \mathcal{U}$, we need only prove that the partial derivative equation 19 is non-positive and the partial derivative equation 20 is non-negative. We will rely heavily on the fact that $f'$ is increasing because $f$ is convex.

We start with equation 19:

$$\frac{\partial}{\partial \tilde{y}_{i \in \mathcal{W}}} d_{\mathcal{Y}}(\tilde{\mathbf{y}}, T) = \begin{cases} 0 & \text{if } \mathcal{S}_{\mathcal{W}} \geq p \text{ and } \mathcal{S}_{\mathcal{U}} \leq q \\ f'\left(\frac{\mathcal{S}_{\mathcal{W}}}{p}\right) - f'\left(\frac{1-\mathcal{S}_{\mathcal{W}}}{1-p}\right) & \text{if } \mathcal{S}_{\mathcal{W}} < p \text{ and } \mathcal{S}_{\mathcal{U}} \leq (1-\mathcal{S}_{\mathcal{W}})\left(\frac{q}{1-p}\right) \\ 0 & \text{if } \mathcal{S}_{\mathcal{U}} > q \text{ and } \mathcal{S}_{\mathcal{W}} \geq (1-\mathcal{S}_{\mathcal{U}})\left(\frac{p}{1-q}\right) \\ f'\left(\frac{\mathcal{S}_{\mathcal{W}}}{p}\right) - f'\left(\frac{1-\mathcal{S}_{\mathcal{W}}-\mathcal{S}_{\mathcal{U}}}{1-p-q}\right) & \text{if } \mathcal{S}_{\mathcal{U}} > (1-\mathcal{S}_{\mathcal{W}})\left(\frac{q}{1-p}\right) \\ & \text{and } \mathcal{S}_{\mathcal{W}} < (1-\mathcal{S}_{\mathcal{U}})\left(\frac{p}{1-q}\right) \end{cases}.$$

Clearly the first and third cases are non-positive, so we proceed to the second case.

Because $\mathcal{S}_{\mathcal{W}} < p$, we have $\frac{\mathcal{S}_{\mathcal{W}}}{p} < 1 < \frac{1-\mathcal{S}_{\mathcal{W}}}{1-p}$ and

$$f'\left(\frac{\mathcal{S}_{\mathcal{W}}}{p}\right) < f'\left(\frac{1-\mathcal{S}_{\mathcal{W}}}{1-p}\right)$$

$$f'\left(\frac{\mathcal{S}_{\mathcal{W}}}{p}\right) - f'\left(\frac{1-\mathcal{S}_{\mathcal{W}}}{1-p}\right) < 0.$$

Next we prove the partial derivative is negative in the fourth case.

$$\mathcal{S}_{\mathcal{W}} < (1-\mathcal{S}_{\mathcal{U}})\left(\frac{p}{1-q}\right)$$

$$\mathcal{S}_{\mathcal{W}} - q\mathcal{S}_{\mathcal{W}} < p - p\mathcal{S}_{\mathcal{U}}$$

$$\mathcal{S}_{\mathcal{W}} - q\mathcal{S}_{\mathcal{W}} - p\mathcal{S}_{\mathcal{W}} < p - p\mathcal{S}_{\mathcal{U}} - p\mathcal{S}_{\mathcal{W}}$$

$$\frac{\mathcal{S}_{\mathcal{W}}}{p} < \frac{1-\mathcal{S}_{\mathcal{U}}-\mathcal{S}_{\mathcal{W}}}{1-p-q}$$

$$f'\left(\frac{\mathcal{S}_{\mathcal{W}}}{p}\right) < f'\left(\frac{1-\mathcal{S}_{\mathcal{U}}-\mathcal{S}_{\mathcal{W}}}{1-p-q}\right)$$

$$f'\left(\frac{\mathcal{S}_{\mathcal{W}}}{p}\right) - f'\left(\frac{1-\mathcal{S}_{\mathcal{U}}-\mathcal{S}_{\mathcal{W}}}{1-p-q}\right) < 0$$

This shows that equation 19 is non-positive and equation 6 is decreasing in $\tilde{y}_i$ for $i \in \mathcal{W}$.

We now consider equation 20:

$$\frac{\partial}{\partial \tilde{y}_{i \in \mathcal{U}}} d_{\mathcal{Y}}(\tilde{\mathbf{y}}, T) = \begin{cases} 0 & \text{if } \mathcal{S}_{\mathcal{W}} \geq p \text{ and } \mathcal{S}_{\mathcal{U}} \leq q \\ 0 & \text{if } \mathcal{S}_{\mathcal{W}} < p \text{ and } \mathcal{S}_{\mathcal{U}} \leq (1-\mathcal{S}_{\mathcal{W}})\left(\frac{q}{1-p}\right) \\ f'\left(\frac{\mathcal{S}_{\mathcal{U}}}{q}\right) - f'\left(\frac{1-\mathcal{S}_{\mathcal{U}}}{1-q}\right) & \text{if } \mathcal{S}_{\mathcal{U}} > q \text{ and } \mathcal{S}_{\mathcal{W}} \geq (1-\mathcal{S}_{\mathcal{U}})\left(\frac{p}{1-q}\right) \\ f'\left(\frac{\mathcal{S}_{\mathcal{U}}}{q}\right) - f'\left(\frac{1-\mathcal{S}_{\mathcal{W}}-\mathcal{S}_{\mathcal{U}}}{1-p-q}\right) & \text{if } \mathcal{S}_{\mathcal{U}} > (1-\mathcal{S}_{\mathcal{W}})\left(\frac{q}{1-p}\right) \\ & \text{and } \mathcal{S}_{\mathcal{W}} < (1-\mathcal{S}_{\mathcal{U}})\left(\frac{p}{1-q}\right) \end{cases}.$$

Clearly the first two cases are non-negative, so we consider the third case.

Because $\mathcal{S}_{\mathcal{U}} > q$, we have $\frac{\mathcal{S}_{\mathcal{U}}}{q} > 1 > \frac{1-\mathcal{S}_{\mathcal{U}}}{1-q}$ and

$$f'\left(\frac{\mathcal{S}_{\mathcal{U}}}{q}\right) > f'\left(\frac{1-\mathcal{S}_{\mathcal{U}}}{1-q}\right)$$

$$f'\left(\frac{\mathcal{S}_{\mathcal{U}}}{q}\right) - f'\left(\frac{1-\mathcal{S}_{\mathcal{U}}}{1-q}\right) > 0.$$

We can no prove the fourth case is positive.

$$\mathcal{S}_{\mathcal{U}} > (1 - \mathcal{S}_{\mathcal{W}})\left(\frac{q}{1-p}\right)$$

$$\mathcal{S}_{\mathcal{U}} - p\mathcal{S}_{\mathcal{W}} > q - q\mathcal{S}_{\mathcal{W}}$$

$$\mathcal{S}_{\mathcal{U}} - p\mathcal{S}_{\mathcal{W}} - q\mathcal{S}_{\mathcal{U}} > q - p\mathcal{S}_{\mathcal{W}} - q\mathcal{S}_{\mathcal{U}}$$

$$\frac{\mathcal{S}_{\mathcal{U}}}{q} > \frac{1 - \mathcal{S}_{\mathcal{W}} - \mathcal{S}_{\mathcal{U}}}{1 - p - q}$$

$$f'\left(\frac{\mathcal{S}_{\mathcal{U}}}{q}\right) > f'\left(\frac{1 - \mathcal{S}_{\mathcal{W}} - \mathcal{S}_{\mathcal{U}}}{1 - p - q}\right)$$

$$f'\left(\frac{\mathcal{S}_{\mathcal{U}}}{q}\right) - f'\left(\frac{1 - \mathcal{S}_{\mathcal{W}} - \mathcal{S}_{\mathcal{U}}}{1 - p - q}\right) > 0$$

This shows that equation 20 is non-negative and equation 6 is increasing in $\tilde{y}_i$ for $i \in \mathcal{U}$.

### 5.2 Alternative Verification Procedure

We developed a second method for using $V$ to verify an amicable perturbation, however we found it more difficult to work with and somewhat less effective and identifying adversarial examples so it was removed from the main paper. Nonetheless, we present it here as an alternative method.

In this method we determine an expected value for $V(\mathbf{x}, \tilde{\mathbf{x}})$ if $M$ has classified $\tilde{\mathbf{x}}$ correctly. We call this expectation $\gamma_{\text{adapt}}$ and reject $\tilde{\mathbf{x}}$ when $V(\mathbf{x}, \tilde{\mathbf{x}}) < \gamma_{\text{adapt}}$ ($V$ indicates that $\tilde{\mathbf{x}}$ is less effective than $M$).

To find $\gamma_{\text{adapt}}$, we interpolate between two points where the expected value of $V$ is known. First, if $d_{\mathcal{Y}}(M(\tilde{\mathbf{x}}), T) = d_{\mathcal{Y}}(M(\mathbf{x}), T)$ (such as when $\mathbf{x} = \tilde{\mathbf{x}}$) we expect no change in the distance to the target set and $\gamma_{\text{adapt}} = 0$. Second, if $d_{\mathcal{Y}}(M(\tilde{\mathbf{x}}), T) = 0$ (implying $M(\tilde{\mathbf{x}})) \in T$) we expect $V(\mathbf{x}, \tilde{\mathbf{x}}) > 0.5$ because we assume $\mathbf{x} \notin T$. We then set $\gamma_{\text{adapt}} = 0.5 + c$ for some confidence parameter $0 < c < 0.5$. (Higher values of $c$ lead to more rejections and more confidence in the $\tilde{\mathbf{x}}$ which pass verification.) Interpolating between the two points yields the formula

$$\gamma_{\text{adapt}} = (0.5 + c) * \left(1 - \frac{\delta_{\mathcal{Y}}(M(\tilde{\mathbf{x}}), T)}{\delta_{\mathcal{Y}}(M(\mathbf{x}), T)}\right). \tag{26}$$

We found that one disadvantage of this technique is you must keep track of the $T$ used to create $\tilde{\mathbf{x}}$. The verification technique used in the paper does not require us to remember this information past creating $\tilde{\mathbf{x}}$.

### 5.3 Additional Implementation Details

In this section we give additional details on how we implemented our methods to create the experimental results found in this paper.

#### 5.3.1 Data Set and Cost Function Details

Here we give additional description of each data set and the corresponding the cost functions $d_{\mathcal{X}}$ used in our experimetns. As noted in Section 3 we must ensure $d_{\mathcal{X}}$ is differentiable. When dealing

with categorical features costs are by nature discrete (and not differentiable). We show how we were able to write these costs in a differentiable form. Suppose $\mathbf{v} \in \mathbb{R}^{\ell}$ is a one-hot encoding of a categorical feature and define the *transition cost matrix* $A$ such that $A_{i,j}$ as the cost of changing from category $i$ to category $j$. Then $\mathbf{z}^T A \tilde{\mathbf{z}}$ represents the costs of changing this categorical feature and is differentiable in $\mathbf{z}$.

**Adult Income**: (Becker & Kohavi, 1996) This widely used data set contains information from the 1994 U.S. census, with individuals labelled by whether their annual income was over \$50,000 ($\sim$\$100,000 in 2023 adjusted for inflation). We define our target set $T$ as over 80% probability high income. Our actionable set allows changes in job type, education and number of hours worked with all other attributes immutable. The cost function $d_{\mathcal{Y}}$ includes the expected number of years to improve education (e.g. two years to go from associate's degree to bachelors degree), a one-year cost to change employer type and the 2-norm of the change in hours worked per week (weighted so 3 hours per week is equivalent to a year spent on education). Here amicable perturbations suggest the best way to improve an individuals odds of making a large income with the least time and effort.

Specifically $d_{\mathcal{X}}$ is the sum cost from changes (1) hours worked per week (2) change in employment type (3) change in education and (4) change in field of work.

The cost from a change in hours is given by $\frac{\Delta h^2}{10}$ where $\Delta h$ is the change in weekly hours worked. This will mean 3 extra hours of work are approximately equivalent to one year of schooling.

The cost from a change in employer (the options are government, private, self-employed and other) is always 1 (equivalent to a year spent on education).

The possible levels of education are (1) any schooling, (2) High School Degree, (3) Professional Degree, (4) some college, (5) Associate's Degree, (6) Bachelors Degree, (7) Master's Degree, (8) Doctorate Degree. The cost transition matrix associated with the level of education (as ordered above) is

$$A_{\text{Education}} = \begin{bmatrix} 0 & 2 & 10 & 3 & 4 & 6 & 8 & 11 \\ L & 0 & 8 & 1 & 2 & 4 & 6 & 9 \\ L & L & 0 & L & L & L & 2 & 5 \\ L & L & 7 & 0 & 1 & 3 & 5 & 8 \\ L & L & 6 & L & 0 & 2 & 4 & 7 \\ L & L & 4 & L & L & 0 & 2 & 5 \\ L & L & 4 & L & L & L & 0 & 3 \\ L & L & 4 & L & L & L & L & 0 \end{bmatrix}, \tag{27}$$

where $L$ is a large number meant to prevent suggestions that lead to a decrease in education, which is impossible (we use $L = 1,000$). These numbers represent the expected number of years required to gain the specified degree (i.e. the cost of going from a high school degree to a bachelors degree is $A_{2,6} = 4$).

Finally the options for fields of work are (1) Service, (2) Sales, (3) Blue-Collar (4) White Collar, (5) Professional, (6) Other. The cost transition matrix associated with the level of education (as ordered above) is

$$A_{\text{Profession}} = \begin{bmatrix} 0 & 1 & 2 & 3 & 4 & 1 \\ 1 & 0 & 1 & 2 & 3 & 1 \\ 1 & 1 & 0 & 1 & 2 & 1 \\ 1 & 1 & 1 & 0 & 1 & 1 \\ 1 & 1 & 1 & 1 & 0 & 1 \\ 1 & 1 & 1 & 1 & 1 & 0 \end{bmatrix}. \tag{28}$$

This represents a cost of 1 for any change

**Law School Success**: (Wightman, 1998) This data set contains demographic information and academic records for over 20,000 law school students labelled by whether or not a student passed the BAR exam. Our target set is an 85% chance of passing the BAR. To create $\mathcal{A}(\mathbf{x})$, we suppose the law school performance is merely a projection that can be changed through more studying, allowing us to change the law school grades and the location where the students take the BAR. The cost function $d_{\mathcal{X}}$ sums the increase in grades and the physical distance travelled to take the BAR where moving to an adjacent region (e.g. Far West to North West) is weighted the same as increasing grades by one standard deviation.

Specifically $d_{\mathcal{Y}}$ sums the increase in grades and the physical distance travelled to take the BAR where moving to an adjacent region (e.g. Far West to North West) is weighted the same as increasing grades one standard deviation. This set up returns the optimal combination of studying harder and moving location to take the BAR. In this data set $d_{\mathcal{X}}$ is sum of the change in grades (in standard deviations from the mean) and distance traveled. The country was divided into eight regions: (1) Far West, (2) Great Lakes, (3) Mid-South, (4) Mountain West, (5) Mid-West, (6) North East, (7) New England, (8) North West. We use the transition cost matrix

$$A_{\text{Region}} = \begin{bmatrix} 0 & 3 & 4 & 1 & 2 & 6 & 5 & 1 \\ 3 & 0 & 1 & 2 & 1 & 2 & 1 & 3 \\ 4 & 1 & 0 & 2 & 1 & 2 & 1 & 5 \\ 1 & 2 & 2 & 0 & 1 & 4 & 3 & 2 \\ 2 & 1 & 1 & 1 & 0 & 3 & 2 & 3 \\ 6 & 2 & 2 & 4 & 3 & 0 & 1 & 5 \\ 5 & 1 & 1 & 3 & 2 & 1 & 0 & 5 \\ 1 & 3 & 5 & 2 & 3 & 5 & 5 & 0 \end{bmatrix} \tag{29}$$

Moves to adjacent regions result in a cost of $1$, while the highest cost of $6$ is incurred by moving from Far West to New England or back.

**Diabetes Prediction**: (for Disease Control & , CDC) This data set contains information on the demographics, health conditions and health habits of 250,000 individuals labelled by whether an individual is diabetic extracted from the Behavioral Risk Factor Surveillance System (BRFSS), a health-related telephone survey that is collected annually by the CDC.. We define $\mathcal{A}(\mathbf{x})$ to allow changes in health habits, BMI, education and income. We use a weighted 2-norm for $d_{\mathcal{X}}$ to represent the relative difficulty of making changes. For example, starting to get regular physical activity is weighted the same as dropping one BMI point. Increasing education, income and health insurance were weighted as more difficult that simply adjusting health habits.

**German Credit**: (Hofmann, 1994) This commonly used data set contains information on 1,000 loan applications in Germany labelled by their credit risk. The actionable set allows for changes in the loan request (time and size) as well as the funds in the applicants checking and savings account and whither the applicant has a telephone. The target set $T$ is a greater than 80% of being a good credit risk. The cost function $d_{\mathcal{Y}}$ is the direct measuring the total difference in Deutsche Marks (DM) between all elements of the application. No cost was assigned to closing empty accounts. The change in length of loan is converted to DM through the individual's monthly disposable income. Finally we set a flat cost of 50DM to acquire a telephone

### 5.3.2 MODEL DETAILS

We used fully connected feed forward neural networks. Each network used 3 hidden layers with ReLu activation functions between each layer. For all data sets except the German Credit data set each hidden layer had $60$ nodes. The German Credit data set required $120$ nodes per layer. Additionally, for the German Credit data set only, we used dropout regularization of $20\%$ on each hidden layer. We trained these models using the ADAM optimizer to minimize cross entropy loss. We used an $80 - 10 - 10$ train-validate-test data split and implemented early stopping with the validation data. All amicable perturbations, counterfactuals and adversarial examples were created for the testing data. We used identical architecture for $V$ as $M$, except for doubling the input size. Accuracy data may be found in table 3.

### 5.3.3 OBJECTIVE FUNCTION DETAILS

In our implementation we formulated the actionablility penalty term $b$ as

$$b(\tilde{\mathbf{x}}) = G \left( \sum_{i=1}^{m} \max\{0, \tilde{\mathbf{x}}_i - u_i\} + \max\{0, l_i - \tilde{\mathbf{x}}_i\} \right) \tag{30}$$

with $G$ a sufficiently large constant.

We formulated our coherence penalty term $p$ as

$$p(\tilde{\mathbf{x}}) = P \sum_{i=1}^{C} \left( 1 - \sum_{j \in \mathcal{C}_i} \tilde{\mathbf{x}}_j \right)^2, \tag{31}$$

with $P$ another appropriately large constant.

The conditioner function $cond$ simply rounded integer and Boolean values to the nearest integer value. For one-hot encoded features categorical features, the category with the largest value set to one and all other categories set to zero.

### 5.3.4 ADDITIONAL RESULTS

Here we show success bar charts similar to those found in figure 7 compare the efficacy of amicable perturbations, counterfactuals Wachter et al. (2017); Mothilal et al. (2020) and adversarial examples from the Carlini Wagner $\ell_2$ attack Carlini & Wagner (2017b) for all data sets. These are similar to Figure 5, but include all data sets and an increased number of cost ($\epsilon$) values.

Each bar chart refers to a particular data set and desired distance $\delta$ to the target set $T$. Inside of each chart, the bars show the percentage of individuals that a method was able to successfully move inside the goal $\delta$ at a variety of costs $\epsilon$. Figure 7 shows data before the verification procedure has been performed and 7 shows the data after all . In these tests, the amicable perturbations (in blue) outperform the counterfactuals (in green and orange) in nearly all cases except for when both methods achieved $100\%$ success or the very high-cost (large $\epsilon$) high reward ($\delta = 0$) scenarios. Carlini Wagner attacks (red) are only effective at larger $\delta$ values because they are designed to move a data point just barely inside the target class. The Carlini Wagner attacks are not required to be actionable (or even feasible), so they do not constitute useful advise. The verifier is able to recognize that these adversarial examples are untrustworthy in all cases.

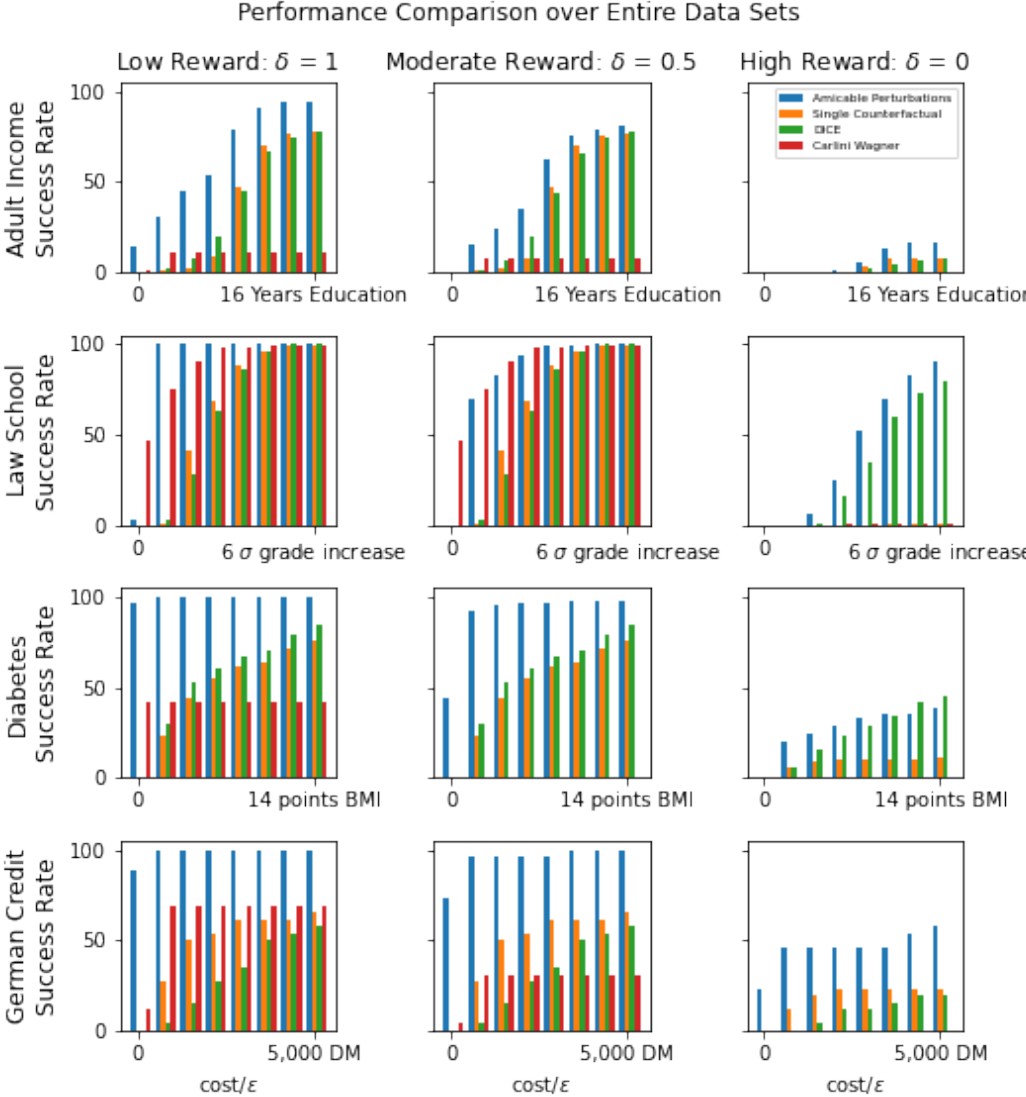

Figure 7: The graphs show average success rate for moving individuals within a variety of distances ($\delta$) to the target set. The y-axis shows the percentage of individuals within the goal distance, and the x-axis, represents different costs ($\epsilon$ values) to achieve the goal. These values were obtained before applying the verification procedure.

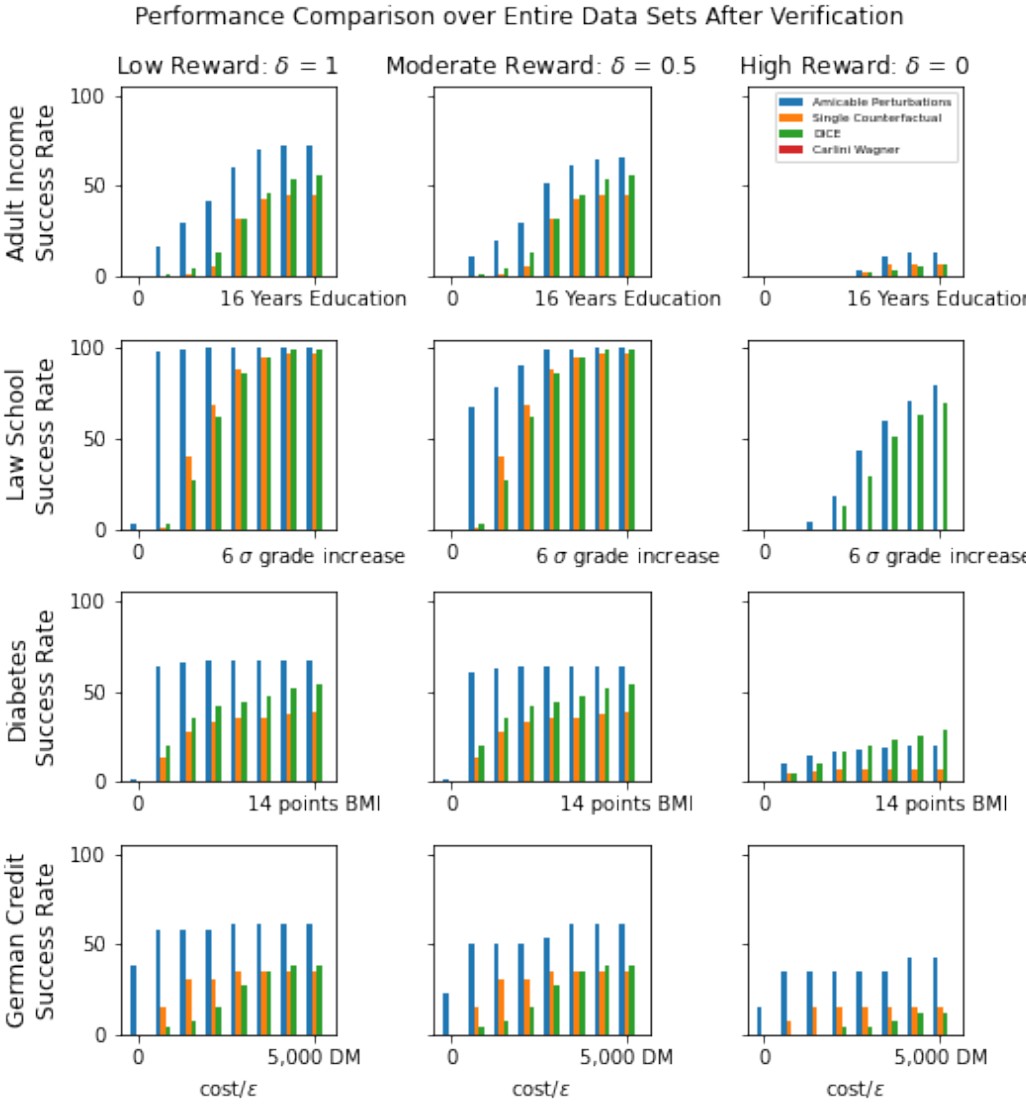

Figure 8: The graphs show average success rate for moving individuals within a variety of distances ($\delta$) to the target set. The y-axis shows the percentage of individuals within the goal distance, and the x-axis, represents different costs ($\epsilon$ values) to achieve the goal. These values were obtained after applying the verification procedure with a $10\%$ chance of eliminating valid inputs.

