# OpenReview forum: "Amicable Perturbations"
_ICLR.cc/2024/Conference — Submitted to ICLR 2024_

### Official Review · Reviewer_cADE · 2023-10-31

**Soundness:** 3 good
**Presentation:** 3 good
**Contribution:** 2 fair
**Rating:** 5
**Confidence:** 3

**Summary:**

"Amicable Perturbations" introduces a novel concept distinct from adversarial examples, guiding users to genuinely alter their data for better outcomes from machine learning classifiers. The paper provides a clear definition, practical scenarios, and a unique verification procedure to ensure authenticity, demonstrating its effectiveness across various datasets and real-world applications.

**Strengths:**

- The paper introduces the concept of "Amicable Perturbations" and provides a comprehensive definition for it. This helps in understanding the potential use cases of amicable perturbations in real-world applications, guiding users to modify their data for better outcomes from machine learning.
- The paper provides a comprehensive verification procedure to ensure the authenticity of the amicable perturbations, which is crucial for real-world applications.

**Weaknesses:**

- The novelty of the paper seems limited. While the concept of amicable perturbations is interesting, it appears to be a minor extension of existing techniques for crafting actionable counterfactual explanations.

- The method's robustness against potential countermeasures is not discussed. What if adversarial training is conducted on the classifier to improve the robustness against adversarial examples? Can the proposed amicable perturbations still work?

- It is unclear whether the verification step really works. The proposed discrepancy score for verification is somewhat heuristic and do not have theoretical guarantees to ensure that the proposed amicable perturbations produce truly different outcomes.

**Questions:**

Is there any theoretical analysis that can explain why the proposed verification procedure can ensure that amicable perturbations produce truly different outcomes?

---

> ### Author Response · Authors · 2023-11-18
> **Responce to Reviewer cADE**
>
> Thank you for your input.  We hope that our comment titled  "Clarification Regarding Novelty of Our Framework" can help explain how our set up differs significantly from previous works on counterfactuals especially in our focus on changing true class probability's instead of merely effecting the classifier's decision (as is the case with most works regarding counterfactual explanations).
>
> With regards to your comment regarding robustness against counter measures: in our set up it is assumed that the entity crafting amicable perturbations is also the creator of the classifier and there is no adversary to deploy countermeasures.  Instead we are focus on ensuring the creator of an amicable perturbation doesn't inadvertently act as an adversary.  If the classifier was made more robust against adversarial attacks, this would increase our confidence in amicable perturbations. Unfortunately the task of creating robust classifiers is very difficult in general and the topic of robust classifiers for tabular data sets specifically is still in its infancy.  We instead seek to leverage our knowledge of both the original ($\mathbf{x}$) and modified ($\tilde{\mathbf{x}}$) data points to device a defence that is not available in the regular adversarial setting.
>
> We hope that our comment "Clarification Regarding the Verifier" can help explain the theory behind how the verifier works and ease your concerns.
>
> Thank you again for your time, and the answers to your specific question is found below.
>
> **Question**: Is there any theoretical analysis that can explain why the proposed verification procedure can ensure that amicable perturbations produce truly different outcomes?
>
> **Answer**: We elaborate on the theory behind the verification method in the comment titled "Clarification on the Verifier."  We also have the numerical justification that it is $100\%$ effective at identifying the adversarial attacks we created for our data set.

---

### Official Review · Reviewer_6MjG · 2023-10-31

**Soundness:** 3 good
**Presentation:** 2 fair
**Contribution:** 2 fair
**Rating:** 3
**Confidence:** 4

**Summary:**

The paper extends the notion of counterfactuals to allow for striking a balance between the effort $\epsilon$ and reward $\delta$. Given an instance $x$, let $\tilde x$ be a modification of it with "effort" defined using some suitable distance $\epsilon = d(x, \tilde x)$. Let $T$ be a collection of some desirable probability distributions (i.e. probability of the true label $p(y|\tilde x)$ satisfies some lower and upper bound constraints). Let reward $\delta$ be some notion of distance between $p(y|\tilde x)$ and the target set $T$. The authors propose a formulation to strike a balance between $\epsilon$ and $\delta$ as opposed to simply using classical methods in either counterfactual research or adversarial examples.

One key novelty is the verification stage in which they train a classifier $f(x, x')$ that predicts if $x$ and $x'$ have the same label. The authors argue that this verification helps in identifying examples that more faithfully reflect changes in the underlying true class, as opposed to the surrogate classifier (like in adversarial examples).

The authors refer to these modifications as "amicable perturbations." They argue that these can be useful, for instance, when the goal is not related to machine learning per se (e.g. for interpretability or robustness research), but, rather, to offer some guidance on how to change the instance such that it belongs to a desired set of classes (e.g. do a minimal change to a given resume in order to increase the probability of landing interviews).

**Strengths:**

- The authors study an interesting setup. They propose an approach for striking a balance between effort and reward and show that their formulation has nice theoretical properties (e.g. differentiable loss).
- The idea of introducing a verification stage is novel as far as I know.

**Weaknesses:**

The primary weakness is the lack of distinction between correlation and causation. The idea of altering an instance $x$ to $\tilde x$ does not really mean that $p(y|\tilde x)$ can be estimated based on the original joint distribution $p(y, x)$. In fact, this is precisely the type of questions the literature in causal analysis focuses on. Amicable perturbation corresponds to what is commonly referred to as do-queries; see for instance Pearl's introduction to causal analysis (https://pubmed.ncbi.nlm.nih.gov/20305706/). The main takeaway is that $p(y|\tilde x)$ (post-intervention) is not necessarily the same distribution as pre-intervention.

But, the main argument in the paper is the claim that one can estimate $p(y| do(x))$ based on the original joint distribution $p(y, x)$, which is wrong. This can even be seen in one of the examples the authors mention in Section 4, in which their approach suggests that an individual should move from the Far West to the Great Lakes in order to improve their chances of passing the BAR exam. In my own opinion, unless this is addressed, the motivation behind the work is questionable.

Second, while the authors distance themselves from counterfactual and adversarial examples, their formulation seems to be an extension of those methods. The authors extend those methods to handle the tradeoff between effort and reward. Compare for example Equations 5, 6, and 7. This by itself is not a major limitation. However, I mention it here because the narrative of the paper suggests that amicable perturbations are quite different, when they aren't. In Figure 4, for instance, counterfactuals lie *along* the Pareto curve for amicable perturbations.

Third, one key novelty in this work is the idea of verification. But, I don't think this is well justified. When a model is trained on pairs of examples $(x, x')$ from the same distribution $\mathcal{D}$, why should the model work well on examples from a different distribution (e.g. $x$ and its perturbation $\tilde x$)? If it could work, then the OOD problem would be solved. There is no empirical evidence that it works in the present paper, aside from the fact that it has some effect.

A few comments about the presentation:
- It would improve readability if $\mathcal{Y}$ is defined from the beginning as a collection of probability distributions. This becomes clear only later in the analysis, which can be confusing.
- Page 2: Missing space in "no greater than q.Then".
- Equation 5: Missing outer parentheses or at least a space should be added after argmin. Currently, the equation reads as if the minimization is applied to the first term only.
- Page 4: typo in "but they but"
- Page 6: typo in "ensure that out solution"
- Page 6: Missing space in "coherent.To solve"
- Page 6: typo in "In Section 2, discussed"
- Page 7: typo in "not be be"
- Page 7: typo in "differentiable and and cannot"

**Questions:**

- Why did the authors present Equations 5 and 6 as separate equations? They seem to be identical except that the notion of distance or loss is instantiated in one case but kept generic in the other.
- Why does Adult income dataset contain 26000 examples only? It should contain over 48,000 examples.
- How is distance $d(x, \tilde x)$ defined in each dataset? For example, the authors claim in Page 8 that one example should be perturbed using "a simple increase in education to the masters level." A masters degree is not a small change.

---

> ### Author Response · Authors · 2023-11-18
> **Responce to Reviewer 6MjG (1/2)**
>
> Thank you for your input.  We appreciate that causality is an important concern in the field of counterfactuals which is gaining more attention.  We believe, however, that the concern of adversarially vulnerable classifiers is also important and worth exploration (as is the goal of this paper).  Although a classifier is accurate in  making predictions on naturally occurring data ($M(\mathbf{x})\approx\mathbb{P}_\mathcal{D}(\mathbf{y}|\mathbf{x})$) it might be inaccurate in making classification on artificially modified data ($M(\tilde{\mathbf{x}}) \not\approx \mathbb{P}_D(\tilde{\mathbf{y}}|\tilde{\mathbf{x}})$) even if $\mathcal{X}$ was created only by adjusting causal features (perhaps based off of a structural causal model) because of weaknesses in the classifier.  More details are found in our comment titled "Clarification Regarding Trustworthiness."  We hope that work in the future will incorporate both the idea of causality and the trustworthiness with respect to adversarial vulnerable models we propose in this paper.
>
> As an aside, statistics about the BAR exam support the suggestion that one should move from the Far West to the Great Lakes in order to improve one's chances of passing the BAR: California has one of the lowest passage rate for the BAR exam at $73.14\%$, whereas states in the Great Lakes Area such as Minnesota or Wisconsin have passage rates of $91.46\%$ and $91.83\%$ respectively (https://lawschooli.com/easiest-bar-exam-to-pass-in-the-us/).
>
> In regards to the novelty of our framework, we acknowledge that our optimization problem (7) is similar to those used by counterfactual and adversarial attack methods (5) and (6) (with a few key differences).  We believe, however, that much of the value and novelty of our framework is found in the clear definition of the problem (Definition 1).  The focus on changing the true probabilities (as a opposed to only focusing on the classifier output) separates us from most works on counterfactuals and all works on adversarial examples.  Although some works on counterfactuals have specified their goal as changing the true class probabilities, we do not believe this problem was well characterized before we introduced the concepts of flexible targets, principled measures of distance to a target and a verification procedure.  More details on this may be found in our comment "Clarification Regarding Framework Novelty."
>
> With regards to our verification procedure, although its is true the verifier $V$ is trained on the same data as $M$, we train $V$ for a different classification problem (to determine if pairs of data are in the same class) and presenting the training data in a different manner (always in pairs).  We suggest this forces $V$ to learn features differently than $M$ which allows for its use in the verification procedure.  More details may be found in our comment "Clarification Regarding the Verifier."
>
> Thank you again for your time, and the answers to your specific questions are found below.
>
> **Question**: Why did the authors present Equations 5 and 6 as separate equations? They seem to be identical except that the notion of distance or loss is instantiated in one case but kept generic in the other.
>
>
> **Answer**: The object of showing Equations 5 and 6 individually was to highlight how the methods used for finding counterfactuals (which evolved from Equation 5) and adversarial examples (which evolved from Equation 6) are extremely similar and can even by identical under certain choices for loss functions and norms.  [1] delves deeper into this idea listing exactly which attacks and counterfactual creation methods can be identical or proven to achieve similar outputs.
>
> **Question**: Why does Adult income data set contain 26000 examples only? It should contain over 48,000 examples.
>
> **Answer**:  We believe this is a result of our data set being cleaned to remove individuals with missing data.

---

> > ### Comment · Reviewer_6MjG · 2023-11-20
> > **Thanks**
> >
> > Thanks a lot for the response.
> >
> > About the first point, if we take (again) the BAR exam example, the fact that 73% of test takers pass the exam in California while 91% pass it in the Great Lakes Area does not say anything about causation. There might turn out to be indeed a causal link in this case (e.g. if the test in California is more difficult) but, in general, this is not always the case. For example, it is possible hypothetically speaking that the test takers in the two states come from different demographic backgrounds (e.g. different age groups) that influence their chances of passing the test (e.g. if older test takers are more knowledgeable), in which case moving to another state would have no influence on the chances of passing the test. My argument is that the method claims statements about causation when it relies solely on correlations and it is known that causal links cannot be inferred from correlations alone.
> >
> > About the second point, I understand that the model $V$ was trained on different task. But, the distribution of instances $p(x)$ is different from the distribution you use at test time. The concern does not really relate to $M$.
> >
> > I hope this clarifies my comments.

---

> ### Author Response · Authors · 2023-11-18
> **Responce to Reviewer 6MjG (2/2)**
>
> **Question**: How is distance $d_\mathcal{X}(\mathbf{x},\tilde{\mathbf{x}})$ defined in each data set? For example, the authors claim in Page 8 that one example should be perturbed using "a simple increase in education to the masters level." A masters degree is not a small change.
>
> **Answer**:  A brief description of $d_\mathcal{X}(\mathbf{x},\tilde{\mathbf{x}})$ in each data set is found on page 7.  The exact function is found on our code.  Based off of reviewer interest we intend on adding more detailed descriptions of the cost functions to the appendix of our revised paper.
>
>   With regards to the specific example listed.  Cost includes the expected number of years to improve ones education ($6$ in this case).  Although this may appear like a large change, it is modest in comparison to the options suggested by a counterfactual method (which suggested over $12$ years of education).  These large changes are simply a result of the original individual being very unlikely to a high income without a significant amount of effort.
>
> [1] Pawelczyk et al (2022), “Exploring Counterfactual Explanations Through the Lens of Adversarial Examples: A Theoretical and Empirical Analysis”, Proceedings of The 25th International Conference on Artificial Intelligence and Statistics (AISTATS)

---

> ### Author Response · Authors · 2023-11-21
> **Responce to Reviewer 6MjG Continued**
>
> Thank you for your response.
>
> Regarding Causality: While we agree with you that taking causality explicitly into account is desirable, but we believe that this does not undermine the contribution of our framework.  In addition to ensuring causality, there are several other challenging issues when attempting to change the true class probabilities of an input which we address in this work; such as the unreliable nature of classifiers on artificially generated data points (please see our detailed comment on Trustworthiness). We should also mention that our framework is flexible enough to incorporate causality constraints as follows: a) the actionable set defined in the paper can be modified to only allow changes to causally related features; b) the verification procedure could also be expanded to include a check for causal relationships. We agree that this is an important and interesting point, and we can certainly add this discussion to the revised version of the paper.
>
> We also note, that following the link regarding rates of passing the BAR  (https://lawschooli.com/easiest-bar-exam-to-pass-in-the-us/) leads to an explanation of the results of a law professor's analysis of the relative difficulty of BAR exams after compensating for the different academic backgrounds of students (original analysis here https://witnesseth.typepad.com/blog/2013/04/more-on-the-most-difficult-bar-exams.html).  In this analysis California (Far West) was the single most difficult BAR exam and the states in the Great Lakes area tended to rank among the easier exams (Wisconsin being the second easiest nationwide).  Although it is true that correlated features without a causal link could lead to some inaccurate results, we feel that this concern does not invalidate the contributions of this paper nor the possibility of useful results through our methods,  as illustrated by our method picking up on the same trend found by an independent analysis of law school data.
>
> Regarding the Verifier:  We appreciate your legitimate point that $\tilde{\textbf{x}}$ may come from a different distribution than the training data, however we hope our detailed comment regarding the classifier explains why we believe our method will be effective at eliminating some ineffective advice.  We fully recognize that data on the results of interventions  (the distribution of $\tilde{\textbf{x}}$) would be very useful.  For the scope of this paper, we did not consider longitudinal data and incorporating  longitudinal data into the verification procedure would be an interesting avenue of future research.  We can certainly include a discussion of this point in the revised paper.
>
> In Summary:  We note that exploration into causal relationships is important and should be the subject of future research.  However, we believe that there are other concerns related to this topic that are also important and neither our paper (nor the vast majority of other works relating to counterfactuals and algorithmic recourse) should be disqualified solely because they focus on these other concerns.
>
> Thank you again for your time spent in reading and responding to our comments, and please let us know if you have any questions.

---

### Official Review · Reviewer_f8ei · 2023-10-31

**Soundness:** 2 fair
**Presentation:** 2 fair
**Contribution:** 2 fair
**Rating:** 5
**Confidence:** 3

**Summary:**

This paper proposes a framework for identifying effective real-world changes, termed "amicable perturbations", that aim to positively influence the classification of data points. Unlike adversarial examples, these perturbations are designed to impact the true class of a data point. The authors introduce a novel method to verify the impact of amicable perturbations on the true class probabilities.


Most importantly, the paper's definition of amicable perturbations is akin to "improving counterfactual explanations" [1]. This problem has originally been identified theoretically in [2] and was conceptualized in [3]. Hence, the claim that "amicable perturbations" are a new concept is probably overstated and unfortunately limits the novelty of the paper's contribution to the field. That being said, the suggested Verifier is still an interesting concept. However, the paper falls short of a more detailed analysis regarding its efficacy and how it theoretically affects the recourse problem. I would evaluate the paper more favourably if the authors instead focused on the analysis of the recourse problem + verifier. For example, the authors could analyze the recourse performance wrt to ground truth label flips and understand the conditions (on the data generating process, classifier performance, etc.) under which successful identification of an "improving counterfactual explanation" that effectively alters the true class of an individual is possible.


-----
References

[1] Freiesleben (2021), "The Intriguing Relation Between Counterfactual Explanations and Adversarial Examples", Minds and Machines

[2] Pawelczyk et al (2022), “Exploring Counterfactual Explanations Through the Lens of Adversarial Examples: A Theoretical and Empirical Analysis”, Proceedings of The 25th International Conference on Artificial Intelligence and Statistics (AISTATS)

[3] Freiesleben et al (2023), “Improvement-Focused Causal Recourse (ICR)”, Proceedings of the AAAI Conference on Artificial Intelligence (AAAI)

**Strengths:**

**Verifier function as a new method to evaluate counterfactual quality**: The most notable strength of the paper is the introduction of the verifier function, which plays a crucial role in the suggested method by assessing whether a generated counterfactual can alter the true underlying label, such as transitioning from high to low credit risk. While the suggested Verifier is an interesting concept, the paper falls short of a more detailed analysis regarding its efficacy. The paper could delve further into the complexity of the verification problem, evaluating the performance, factors influencing it, and the conditions indicating successful identification of an "improving counterfactual explanation" that effectively alters the true class of an individual.

**Weaknesses:**

**Contribution**: The paper overstates its contribution to the existing literature, neglecting to establish connections with the counterfactual explanation literature, which has already highlighted the concept that represents "amicable perturbations" in previous works, such as [1-3]. Further, the problem of controlling a counterfactuals classification confidence has also been addressed in works that deal with generating robust recourse (see [4,5]).

**Evaluation of the Verifier**: The evaluation of the Verifier using real-world data poses a significant challenge, given the absence of empirical evaluations to ascertain its accuracy in identifying improving counterfactuals. I recommend incorporating Structural Causal Models (SCMs) from the causal literature to comprehensively validate the efficacy of the proposed Verifier. Moreover, exploring the feasibility of an end-to-end optimization that uses the Verifier as constraint in the optimization could potentially enhance the study's robustness and applicability.

As a side note: The verifier is differentiable (or can be made differentiable) as far as I see. One could potentially consider an end-to-end optimization of this problem.

-----
**References**

[1] Freiesleben (2021), "The Intriguing Relation Between Counterfactual Explanations and Adversarial Examples", Minds and Machines

[2] Pawelczyk et al (2022), “Exploring Counterfactual Explanations Through the Lens of Adversarial Examples: A Theoretical and Empirical Analysis”, Proceedings of The 25th International Conference on Artificial Intelligence and Statistics (AISTATS)

[3] Freiesleben et al (2023), “Improvement-Focused Causal Recourse (ICR)”, Proceedings of the AAAI Conference on Artificial Intelligence (AAAI)

[4] Dominguez-Olmedo et al (2022), “On the Adversarial Robustness of Causal Algorithmic Recourse” Proceedings of the 39-th International Conference on Machine Learning (ICML)

[5] Pawelczyk et al (2022), “Probabilistically Robust Recourse: Navigating the Trade-offs between Costs and Robustness in Algorithmic Recourse”, International Conference on Learning Representations (ICLR)

**Questions:**

Please see above.

---

> ### Author Response · Authors · 2023-11-18
> **Responce to Reviewer f8ei**
>
> Thank you for your input.  We are aware that other works have considered the relationship between adversarial examples and counterfactuals, however, we believe that without our framework the task of changing true class probabilities (and avoiding the accidental creation of adversarial examples) has not been well characterized.  More explanation may be found in our comment "Clarification Regarding Framework Novelty."  In the comment titled "Clarification regarding Trustworthiness" we explain how our goals and techniques differ from those of [4] and [5].  In our revised paper we more clearly compare and contrast our work to previous contributions in the field of counterfactuals.
>
> We avoid incorporating the verifier into the optimization problem (to form an end-to-end optimization problem) Because we believe this would enable the optimizer to find inputs that are adversarial to both the Classifier $M$ and verifier $V$.  Please see our comment "Clarification regarding the Verifier" for a more detailed explanation.

---

### Official Review · Reviewer_v88R · 2023-11-02

**Soundness:** 3 good
**Presentation:** 3 good
**Contribution:** 2 fair
**Rating:** 5
**Confidence:** 3

**Summary:**

The authors propose the problem setup of _amicable perturbations_, in which the goal is to generate a realistic perturbation to a person's features in a way that causes a desired change in the classifier prediction. To generate amicable perturbations, the authors propose an optimization problem to minimize the weighted sum of the cost of the perturbation and the distance to the desired target. To avoid generating adversarial samples, the authors propose a second verification step to filter out such samples. The authors evaluate their method on several tabular datasets, finding that they can generate perturbations matching certain problem specifications at a higher success rate than the baselines.

**Strengths:**

- The paper tackles an important machine learning problem.
- The paper is generally well-written and easy to understand.

**Weaknesses:**

1. The perturbations generated by the method are not _causal_, unlike previous work [e.g. 1-3]. As such, if the classifier learns some spurious correlations, the proposed method could suggest perturbations that do not causally lead to changes in the true label.

2. The authors frame their problem of amicable perturbations as an entirely new problem setting. This is an oversell in my opinion, as the proposed setup is essentially the same as a standard algorithmic recourse setup, with the main novelties being that (1) the authors generalize the target set instead of a simple label flip, and (2) the authors propose the second verification step. Note that the idea of adversarially robust recourse has been explored in several prior work [2, 3] which the authors have not referenced.

3. The scenario of what to do when the verifier rejects a candidate perturbation is underexplored in the paper. The authors propose some actions at end of Section 3, but I don't believe these were used in the experiments. I would be particularly interested to see whether action (2) can help in raising the success rate.

4. The authors should show a few cases of amicable perturbations rejected by the verifier, to visually confirm that these are indeed adversarial examples.

5. The authors should conduct an ablation study on e.g. the addition of b(x) and p(x), and the choice of the f-divergence.

6. The authors formulate their target set $T$ to be quite general, but then only test on binary classification datasets. They should consider adding some multi-class classification datasets.


[1] Algorithmic recourse under imperfect causal knowledge: a probabilistic approach. NeurIPS 2020.

[2] On the Adversarial Robustness of Causal Algorithmic Recourse. ICML 2022.

[3] Probabilistically Robust Recourse: Navigating the Trade-offs between Costs and Robustness in Algorithmic Recourse. ICLR 2023.

**Questions:**

Please address the weaknesses above, and the following questions:

1. Have the authors considered a projected gradient descent based algorithm, instead of the addition of the b(x) and p(x) penalties?

2. For the optimization problem in Eq (7), since the f-divergence is 0 once $M(\tilde{x}) \in T$, it seems like all generated perturbations should be right on the boundary of $T$. However, this is not what happens in practice. Can the authors give some intuition on this?

3. For the definition of $\Delta(x, \tilde{x})$, it seems to me that the threshold should be dependent on T (i.e. $p$ and $q$). Can the authors give some intuition on why they use a fixed threshold $\gamma$?

4. In Algorithm 1, have the authors considered adding an early stopping criteria based on $\epsilon$ and $\delta$ to the first for loop?

---

> ### Author Response · Authors · 2023-11-18
> **Responce to Reviewer v88R (1/2)**
>
> Thank you for your input.  We appreciate your concern about causal relations and acknowledge that this is an interesting avenue for future research involving our framework and techniques.  This paper focuses on the separate but equally important concern that modified data points may not produce the anticipated result because ML classifiers are often very inaccurate on modified data point (such as adversarial attacks).  Please see our comment titled "Clarification Regarding Trustworthiness" for more details about this distinction.  This comment also helps explain how our goal is fundamentally different from the goals of [2] and [3].  We also hope the comment titled "Clarification Regarding Trustworthiness" will address some of your concerns regarding the relationship between our paper an previous works.
>
> Thank you again for your time, and the answers to your specific questions are found below.
>
> **Question**: Have the authors considered a projected gradient descent based algorithm, instead of the addition of the $b(\mathbf{x})$ and $p(\mathbf{x})$ penalties?
>
> **Answer**: Yes.  We initially tried projected gradient descent, but we found that if we projected after every step of the gradient descent our output never changed.  If we projected only after the gradient descent had converged, we found that the results were very far from our target (sometimes even further than the original input).  We determined this was because the path of the gradient descent went far outside the actionable set and made highly incoherent inputs (i.e. having multiple values $\gg 1$ in what should be a one-hot encoding).  This often meant that the final projection step erased all progress made my the gradient descent.
>
> We determined it would not be feasible to search for the proper number of steps between projections for each individual optimization task, and instead we adopted the use of $b(\tilde{\mathbf{x}})$ and $p(\tilde{\mathbf{x}})$.
>
> **Question**: For the optimization problem in Eq (7), since the f-divergence is 0 once $M(\tilde{\mathbf{x}})\in T$, it seems like all generated perturbations should be right on the boundary of $T$. However, this is not what happens in practice. Can the authors give some intuition on this?
>
> **Answer**:  That is a very astute observation.  We had initially thought it might be necessary to run the optimization on a slightly smaller target set in order push the amicable perturbations inside the border of $T$.  In practice, however, we found that even though the $\tilde{\mathbf{x}}$ produced immediately after gradient descent lied on the border of $T$, projecting $\tilde{\mathbf{x}}$ onto the coherent space ($cond(\tilde{\mathbf{x}})$) had a tendency to move the amicable perturbation inside of $T$.  It is worth noting that even though the amicable perturbations we produce often lie inside of $T$, we do not produce "overkill" examples that are far inside the border of $T$.  The example counterfactual from the Law School Success data set in page 8 illustrates how other methods do exhibit this "overkill" behavior.
>
> **Question**: For the definition of $\Delta(\mathbf{x},\tilde{\mathbf{x}}))$, it seems to me that the threshold should be dependent on T (i.e. $p$ and $q$ ). Can the authors give some intuition on why they use a fixed threshold $\gamma$?
>
> **Answer**:  We initially considered two options for the verification procedure: the one found in our paper and a procedure were we compared $V(\mathbf{x},\tilde{\mathbf{x}})$ to an adaptive threshold ($\gamma_{\text{adapt}}$) determined by the $\delta$ values of $\mathbf{x}$ and $\tilde{\mathbf{x}}$ (which are dependent on $p$ and $q$).  We defined the adaptive threshold as
> $$
> \gamma_{\text{adapt}}  = (0.5 + c)*\left (1 - \frac{\delta_{\mathcal{Y}}(M(\tilde{\mathbf{x}}),T)}{\delta_{\mathcal{Y}}(M(\mathbf{x}),T)}\right ).
> $$
>  We came to this formula by interpolating between two point where the expected value of $V$ is known. First, if $d_\mathcal{Y}(M(\tilde{\mathbf{x}}),T) = d_\mathcal{Y}(M({\mathbf{x}}),T)$ (such as when $\mathbf{x}$ = $\tilde{\mathbf{x}}$) we expect no change in the distance to the target set and $\gamma_{\text{adapt}} = 0$.  Second, if $d_\mathcal{Y}(M(\tilde{\mathbf{x}}),T) = 0$ (implying $M(\tilde{\mathbf{x}}))\in T$) we expect $V(\mathbf{x},\tilde{\mathbf{x}}) > 0.5$ because we assume $\mathbf{x}\notin T$.  We then set $\gamma_{\text{adapt}} = 0.5+c$.
>
>  In practice we found that the method presented in our paper was both easier to work with and more effective at eliminating adversarial examples.  Running into paper length constraints, we decided to cut the "adaptive threshold method."  We can certainly add this additional discussion in the Appendix/Supplementary material. For more explanation of the verification method in our paper please see our comment title "Clarification on the Verifier."

---

> ### Author Response · Authors · 2023-11-18
> **Responce to Reviewer v88R (2/2)**
>
> **Question**: In Algorithm 1, have the authors considered adding an early stopping criteria based on $\epsilon$ and $\delta$ to the first for loop?
>
>  **Answer**:  An early stopping criterion based on convergence would be useful to decrease computational time.  We  believe this criterion should be based on convergence, not $\epsilon$ and $\delta$, because we wish to find the optimal cost-reward trade off.  A stopping criterion based off of $\epsilon$ and $\delta$ would instead find the first $\tilde{\mathbf{x}}$ on the optimization path that satisfies an $\epsilon,\delta$ requirement.

---

### Official Review · Reviewer_syxy · 2023-11-15

**Soundness:** 4 excellent
**Presentation:** 3 good
**Contribution:** 4 excellent
**Rating:** 8
**Confidence:** 3

**Summary:**

The paper proposes a novel type of input perturbations termed amicable perturbations whose goal is to find the most efficient changes to an input in order to achieve a more favorable classification outcome in the real world. The efficiency of input changes is quantified using a cost or distance function that could be domain (dataset) dependent, and the changes are constrained to be realistic/meaningful. They clearly contrast amicable perturbations from counterfactual and adversarial inputs, and provide a number of real-world examples to motivate it. Different from adversarial inputs, amicable perturbations seek to modify the true class of an input.

The paper then defines the idea of an $(\epsilon, \delta)$ amicable perturbation, where $\epsilon$ corresponds to the cost of making the change from $x$ to $\tilde{x}$ and $\delta$ corresponds to how closely we want the true class of $\tilde{x}$ to approach the target desirable set. They propose a closed-form solution for the statistical distance to the target set for a family of f-divergence based distances (Theorem 1), which is an interesting result. Based on this, they propose a constrained optimization algorithm to find amicable perturbations for neural network classifiers on tabular datasets.

**Strengths:**

A novel and principled framework for creating optimal changes to an input in the most efficient way in order to achieve a desirable outcome in the real world. Clearly discusses and contrasts amicable perturbations with counterfactual inputs and adversarial perturbations.

The changes suggested by amicable perturbations offer useful insights (advice) that can help people or products in real world situations. This is discussed with many examples and also shown in the experiments. Therefore, the intended use case of amicable perturbations is for enabling social improvement.

The development of the formalism in terms of the actionable set, effort/cost function, and the desired goal is very principled. The analysis of the statistical distance to the target set in terms of f-divergence and deriving its closed form solution as a piecewise linear function, which is continuously differentiable (Theorem 1) is a neat contribution.

Overall, the paper presents an interesting and novel contribution, which could spur a new line of research similar to adversarial examples.

**Weaknesses:**

**1.** The proposed amicable perturbations are mainly suitable for tabular datasets, with well-defined input features. It is not clear how they can be applied to image and text datasets. A discussion on this would be useful.

**2.** The method supports only differentiable models such as deep neural networks, but cannot be applied to non-differentiable models such gradient-boosted decision trees or random forest. However, the latter models are widely used for tabular data since they often have better performance. The ability to handle such models would provide more flexibility to their proposed method.

**3.** Some of the details in Algorithm 1 are not clear. For instance, how is the projection function to satisfy coherency $cond(\tilde{x})$ defined? Since the features of the tabular datasets are often categorical or binary, it seems like using gradient descent is not the best approach to optimizing the perturbation. They have to introduce penalty terms to remain in the actional set $\mathcal{A}(x)$, and a projection function $cond(x)$ to satisfy coherency. There is not enough discussion on how effective this approach is and its limitations. I would suggest including a paragraph on the limitations and future work (scope for improvement) covering these aspects.

**4.** The paper applies the Carlini-Wagner $\ell_2$ attack to generate adversarial inputs. However, it does not seem appropriate to apply this attack for these tabular datasets where some of the features are categorical or binary. It is likely to find adversarial inputs that are easily detectable by the verification function.

**5.**  Minor: needs some proof-reading for language and typos. Some details about the algorithm are missing and could be included. Please see my suggestions in the `Questions` section.

**Questions:**

**1)** The citation format of references is not correct. There should be a parentheses around citations; e.g. (Leo et al., 2019) instead of Leo et al., 2019.

**2)** The paper needs some proofreading for language issues and typos. For instance, in the last paragraph of page 2, it should be:
> “In Section 2, we define an amicable perturbation as well as contrast it with related work.”

**3)** The paper needs to define $k$ as the number of classes and $m$ as the input dimension at the start of Section 2. Sometimes these two symbols are interchanged leading to confusion. A few instances are listed below:
- In Eqn 28, it should be $m$ instead of $k$ denoting the number of features.
- In the proof of theorem 1 (Appendix 5.1), the number of classes is denoted by $m$ instead of $k$.
- On page 6, under `Solving Step 1`, the projection function should be defined as $\text{cond} : \mathrm{R}^m \mapsto \mathcal{X}$.

### 4) Section 2, Problem setting and goals
The notations could be more clear. Rather than saying $(\mathcal{X}, \mathcal{Y}) \sim \mathcal{D}$, it would be better to say $(x, y) \sim \mathcal{D}$, where $x \in \mathcal{X} \subset \mathrm{R}^m$ and $y \in \mathcal{Y}$. Here, $\mathcal{Y}$ is the $k$ probability simplex.

Also, different from conventional notation where $y$ is an integer, here it is the true (conditional) probability distribution over the set of classes. Therefore, it would be clearer to define a class random variable $C \in \\{1, \cdots, k\\}$ and $y := [P(C=1 | x), \cdots, P(C=k | x)]$, i.e. the true class posterior probabilities. Of course, this could specialize to a one-hot coded label.

With this definition, it is clear that an amicable perturbation $\tilde{x}$ aims to change the true class of an input to a desired $\tilde{y} := [P(C=1 | \tilde{x}), \cdots, P(C=k | \tilde{x})]$.

In the definition of the difference training data (before Eqn 3), used to train $V(x, \tilde{x})$, it should not include pairs $(i, j)$ where $i = j$. If the architecture of $V$ does not depend on the order of inputs, then it should only include pairs where $j > i$.

In Eqn (3), it would be simpler to define $z^{(i,j)} = \mathrm{1}[y^{(i)} = y^{(j)}]$, i.e. using the indicator function.

### 5) On training the verification function (page 4)
The verification function $V$ is actually estimating the conditional probability that the true class corresponding to $x$ and $\tilde{x}$ are equal given the inputs. Let $C$ and $\tilde{C}$ denote the true class corresponding to $x$ and $\tilde{x}$. The verification function estimates the probability $P(C = \tilde{C} | x, \tilde{x})$. Another way to estimate this probability is using the classifier $M(x)$, assuming conditional independence of $C$ and $\tilde{C}$ given $(x, \tilde{x})$, as follows:
$\hat{P}(C = \tilde{C} | x, \tilde{x}) = \sum_{i=1}^k M_i(x) M_i(\tilde{x})$

**6)** For solving step 2 (page 6), it is mentioned that a large random sample of input pairs from the test set which have different labels are used to set the threshold $\gamma$. Is it appropriate to use the labeled test set for deciding the verification threshold without introducing bias?

### 7) Regarding Algorithm 1
Would suggest defining the penalty terms $b(\tilde{x})$, $p(\tilde{x})$ and the projection function $cond(x)$. Is it possible to provide a general form for different scenarios? It would be good to provide pointers in the main paper to the penalty terms defined in Appendix 5.2.1.

In the step where the gradient $\textbf{g}$ is calculated, please show that the gradient is over all the terms by including parentheses around the terms.

In practice, is it sufficient to use a fixed learning rate $\alpha$ for the gradient descent? Have the authors explored adaptive learning rate schemes?

Please add comments for some of the lines in the algorithm for clarity. For example, can add a comment like “// Projection to ensure coherancy” to the line $\tilde{x} = cond(\tilde{x})$.

It would help to provide a precise method (recipe) to adjust $\lambda$ and the problem parameters. What is the method used in your implementation?

**8)** It would be interesting to explore non-gradient based optimization methods for finding the amicable perturbations. This would allow us to extend the method to non-differentiable models such as gradient-boosted decision trees.

**9)** In the experimental setup (Section 4), the cost function for each dataset should be $d_{\mathcal{X}}$, not $d_{\mathcal{Y}}$. Same comment for Figure 3.

**10)** In Section 4, the cost functions $d_{\mathcal{X}}$ defined for the datasets are somewhat vague. Could the authors describe them more precisely in an appendix?

**11)** For the plots in Figure 4, how do we interpret the scale of $\epsilon$ values on the x-axis? In the discussion in the text for the Law School dataset, it is mentioned that a short move from the Far West to the Great Lakes region and a mild increase in grades can result in a 11% increase. It is hard to understand how to translate these changes into the $\epsilon$ (effort) values. Similar comment for the other examples.

**12)** The following lines under `Other Methods` in Section 4 is not clear.
> The counterfactuals will belong to the same actionable set as the amicable perturbations, but the adversarial examples not be be in the actionable set or even to be coherent. This is significant because our verifier procedure should be able to recognize that these adversarial examples are not effective real world solutions.

**13)**  More details on the neural networks used for each dataset in the appendix would be useful. How are they made suitable for handling categorical inputs?

**14)**  Minor: the quality of math symbols in the figures can be improved.

**15)**  A discussion of limitations and societal impacts of amicable perturbations would be useful. Can include points such as the computational cost of generating amicable perturbations, which may not be so bad in the real world because it is not usually time sensitive. The verification method for detecting inputs could potentially be improved to reduce the false rejection of amicable perturbations as adversarial.

**16)**  Is it possible to effectively apply amicable perturbations to domains such as image and text? Could you discuss some potential use cases?

---

> ### Author Response · Authors · 2023-11-18
> **Responce to Reviewer syxy (1/3)**
>
> Thank you for your input.  We will look into how are method could be applied to non-tabular data sets as this is an avenue we had not considered.
>
> In regards to your comments on expanding our methods to non-differentiable models, we believe this is a promising avenue for future research.  Many works in counterfactual explanations have sought to find improved methods for solving the mixed integer programming problem (usually involving genetic algorithms or decision  tree specific methods).  We hope that these methods may be adapted to our framework.
>
> We are working to improve the explanation of Algorithm 1 in our revised version.
>
> We understand your concerns about the use of the Carlini-Wagner attack for a tabular data set, however, the field of adversarial attacks on tabular data is in its infancy with seminal work [1].  We were unable to find any trusted implementation of an adversarial attack on tabular data, so we decided that Carlini-Wagner was our best option.
>
>
>
> We are very grateful for your help and suggestions for improving language and catching typos.
>
> Thank you again for your time, and the answers to your specific questions are found below.
>
> **Question**: The citation format of references is not correct. There should be a parentheses around citations; e.g. (Leo et al., 2019) instead of Leo et al., 2019.
>
> **Answer**:  You are correct.  We are correcting that error in our updated version.
>
> **Question**: The paper needs some proofreading for language issues and typos. For instance, in the last paragraph of page 2, it should be:
>
>     “In Section 2, we define an amicable perturbation as well as contrast it with related work.”
>
>
> **Answer**:  You are correct.  We are currently proof reading our paper to improve readability and catch typos.  We have corrected the error you pointed out.
>
> **Question**:  The paper needs to define $k$ as the number of classes and $m$ as the input dimension at the start of Section 2. Sometimes these two symbols are interchanged leading to confusion. A few instances are listed below:
>
> -In Eqn 28, it should be $m$ instead of $k$ denoting the number of features.
>
> -In the proof of theorem 1 (Appendix 5.1), the number of classes is denoted by $m$ instead of $k$.
>
> -On page 6, under Solving Step 1, the projection function should be defined as $cond:\mathbb{R}^m\rightarrow\mathcal{X}$.
>
> **Answer**:  Thank you for catching those errors.  We have made those corrections in our updated paper.
>
> **Question**:  Section 2, Problem setting and goals
>
> The notations could be more clear. Rather than saying $(\mathcal{X},\mathcal{Y})\sim \mathcal{D}$, it would be better to say $(\mathbf{x},\mathbf{y})\sim \mathcal{D}$, where $\mathbf{x}\in\mathcal{X}\subset\mathbb{R}^m$ and $\mathbf{y}\in\mathcal{Y}$. Here, $\mathcal{Y}$ is the $k$ probability simplex.
>
> Also, different from conventional notation where $\mathbf{y}$ is an integer, here it is the true (conditional) probability distribution over the set of classes. Therefore, it would be clearer to define a class random variable $C\in\{1,...,k\}$ and $\mathbf{y}:=[\mathbb{P}(C=1|\mathbf{x}),...,\mathbb{P}(C=k|\mathbf{x})]$, i.e. the true class posterior probabilities. Of course, this could specialize to a one-hot coded label.
>
> With this definition, it is clear that an amicable perturbation $\tilde{mathbf{x}}$ aims to change the true class of an input to a desired $\tilde{\mathbf{y}}:=[\mathbb{P}(C=1|\tilde{\mathbf{x}}),...,\mathbb{P}(C=k|\tilde{\mathbf{x}})]$.
>
> In the definition of the difference training data (before Eqn 3), used to train $V(\mathbf{x},\tilde{\mathbf{x}})$, it should not include pairs $(i,j)$ where $i=j$. If the architecture of $V$ does not depend on the order of inputs, then it should only include pairs where $i>j$.
>
> In Eqn (3), it would be simpler to define $z^{(i,j)} = 1[\mathbf{y}^{(i)} = \mathbf{y}^{(j)}]$, i.e. using the indicator function.
>
> **Answer**:  We appreciate your suggestions for improving clarity. We are looking into updating those notations for clarity while still maintaining consistent notations with other papers in this field.
>
> **Question**:  The verification function is actually estimating the conditional probability that the true class corresponding to $\mathbf{x}$ and $\tilde{\mathbf{x}}$ are equal given the inputs.  Let $C$ and $\tilde{C}$ denote the true class corresponding  to $\mathbf{x}$ and $\tilde{\mathbf{x}}$. The verification function estimates the probability $\mathbb{P}(C,\tilde{C}|\tilde{\mathbf{x}})$. Another way to estimate this probability is using the classifier $M(\mathbf{x})$, assuming conditional independence of $C$ and $\tilde{C}$ given $(\mathbf{x},\tilde{\mathbf{x}})$, as follows: $\hat{\mathbb{P}}(C,\tilde{C}|\tilde{\mathbf{x}},\mathbf{x}) = \sum_{i=1}^k M_i(\mathbf{x})M_i(\tilde{\mathbf{x}})$
>
> **Answer**:  We are trying to make this explanation more clear in our revised version and appreciate your suggestions on how we could do this.

---

> ### Author Response · Authors · 2023-11-18
> **Responce to Reviewer syxy (2/3)**
>
> **Question**:  For solving step 2 (page 6), it is mentioned that a large random sample of input pairs from the test set which have different labels are used to set the threshold $\gamma$. Is it appropriate to use the labeled test set for deciding the verification threshold without introducing bias?
>
> **Answer**:  We use cross validation testing data which has not been used to train the classifier.  We hope limits that amount of biased this could add.  It would be preferable to use previously created amicable perturbations that had been tested in the real world (in order to assign labels), but we don't assume we have access to that kind of data.
>
> **Question**:   Regarding Algorithm 1
>
> Would suggest defining the penalty terms $b(\tilde{\mathbf{x}})$, $p(\tilde{\mathbf{x}})$ and the projection function $cond(\tilde{\mathbf{x}})$. Is it possible to provide a general form for different scenarios? It would be good to provide pointers in the main paper to the penalty terms defined in Appendix 5.2.1.
>
> In the step where the gradient $\mathbf{g}$ is calculated, please show that the gradient is over all the terms by including parentheses around the terms.
>
> In practice, is it sufficient to use a fixed learning rate $\alpha$ for the gradient descent? Have the authors explored adaptive learning rate schemes?
>
> Please add comments for some of the lines in the algorithm for clarity. For example, can add a comment like “// Projection to ensure coherency” to the line $\tilde{\mathbf{x}} = cond(\tilde{\mathbf{x}})$.
>
> It would help to provide a precise method (recipe) to adjust
> and the problem parameters. What is the method used in your implementation?
>
> **Answer**:  Thank you for your feedback.  We are attempting to clarify our description of Algorithm 1 based off of your recommendations.  In practice we used the ADAM modification of gradient descent.  We will mention that fact in the updated version.
>
> **Question**: It would be interesting to explore non-gradient based optimization methods for finding the amicable perturbations. This would allow us to extend the method to non-differentiable models such as gradient-boosted decision trees.
>
> **Answer**:  We agree that this is an interesting avenue for future research.
>
> **Question**: In the experimental setup (Section 4), the cost function for each data set should be $d_\mathcal{X}$, not $d_\mathcal{Y}$. Same comment for Figure 3.
>
> **Answer**:  You are correct.  We have corrected those errors in our updated version.
>
> **Question**: In Section 4, the cost functions $d_\mathcal{X}$ defined for the data sets are somewhat vague. Could the authors describe them more precisely in an appendix?
>
> **Answer**: We are adding a section to the appendix that includes more details on the cost functions.  We also note that the exact code for the cost functions is found in the supplementary material.
>
> **Question**: For the plots in Figure 4, how do we interpret the scale of $\epsilon$ values on the x-axis? In the discussion in the text for the Law School data set, it is mentioned that a short move from the Far West to the Great Lakes region and a mild increase in grades can result in a $11\%$ increase. It is hard to understand how to translate these changes into the $\epsilon$ (effort) values. Similar comment for the other examples.
>
> **Answer**: We are adding cost values to the x-axis in figure 4 similar to those in bar charts from figure 5.
>
> **Question**: The following lines under Other Methods in Section 4 is not clear.
>
>     The counterfactuals will belong to the same actionable set as the amicable perturbations, but the adversarial examples not be be in the actionable set or even to be coherent. This is significant because our verifier procedure should be able to recognize that these adversarial examples are not effective real world solutions.
>
>
> **Answer**: We will clarify those lines in the updated version.  They are meant to explain that we have the same actionable set requirements for the counterfactuals as we do for the amicable perturbations.  The Carlini-Wagner attack, however, does not allow for the limiting of the set within which the adversarial examples are found.  Hence the adversarial examples may lie outside the actionable set and and also break coherency (formatting) rules.
>
> **Question**:  More details on the neural networks used for each data set in the appendix would be useful. How are they made suitable for handling categorical inputs?
>
>
> **Answer**: We are adding a section on classifier details to the appendix.
>
> **Question**:  Minor: the quality of math symbols in the figures can be improved.
>
>
> **Answer**: We will look into ways to decrease pixelation.

---

> ### Author Response · Authors · 2023-11-18
> **Responce to Reviewer syxy (3/3)**
>
> **Question**: A discussion of limitations and societal impacts of amicable perturbations would be useful. Can include points such as the computational cost of generating amicable perturbations, which may not be so bad in the real world because it is not usually time sensitive. The verification method for detecting inputs could potentially be improved to reduce the false rejection of amicable perturbations as adversarial.
>
> **Answer**:  We will include those ideas in the conclusion of our revised paper.
>
> **Question**:  Is it possible to effectively apply amicable perturbations to domains such as image and text? Could you discuss some potential use cases?
>
> **Answer**:  This is an interesting question for future research.  This will be very dependent on the context of the problem and there will have to be a definable actionable set.  Perhaps such an actionable set for image data could be: you are allowed to change color saturation and contrast.  We also think it would be more useful if the classification task were something like "is the image pleasing or not", as opposed to trying to identify the subject of the image.
>
>
>
> [1] Ballet, V., Renard, X., Aigrain, J., Laugel, T., Frossard, P., & Detyniecki, M. (2019). Imperceptible adversarial attacks on tabular data. arXiv preprint arXiv:1911.03274.

---

### Author Response · Authors · 2023-11-18
**Clarification Regarding Framework Novelty**

We suspect there was some confusion with reviewers believing that the new framework we presented was entirely encapsulated by the optimization problem (7), which is indeed similar previous counterfactual set ups (with a few key distinctions).  We would argue that our framework is much more than just that optimization problem and is better encapsulated by Definition 1.  Since counterfactuals were first proposed by [1] as a way to explain a classifiers logic they have been repurposed for many tasks under a variety of names (i.e. actionable counterfactual, algorithmic recourse, improvement-focused causal-recourse, etc..) often including an new optimization problem or methodology, but seldom containing a clearly defined goal and purpose.  This led to [2], which is entirely focused on putting forward definitions for the different counterfactual related tasks.  We believe that clearly defining our task and establishing the supporting framework is important  and that our problem set up is both useful and novel for a variety of reasons:

- We clearly define the goal as affecting the true class distribution instead of the classifier output (which makes the verification step necessary).  To understand this distinction consider that the tasks of  (a) convincing an ML classifier to offer you a loan and (b) preparing one's self to be able to pay off a loan.  It is possible to complete task (a) without completing task (b) by simply "fooling the classifier" We are aware of only one other paper [3] that clearly defines this as their task.  This paper does not, however, include several additional problem parameters (listed in the bullet points below) we see as necessary when using this goal.
- A flexible target set is (to our knowledge) a completely novel idea in this field.  This is essential when seeking to change ground truth probabilities.  Otherwise one is simply requiring a $51\%$ probability of being in the desirable class which is unlikely to be a useful target in most cases.
- The idea of defining the distance to our target in a mathematically principled way is also entirely novel (to our knowledge).   This is necessary  when the goal is flexibly defined instead of merely "flipping the label."  Additionally our definition of this distance (in combination with the target set) allows us to expand to multi-class settings which is lacking in most counterfactual papers. Our theoretical result stated in Theorem $1$ gives a neat characterization of this loss, for a very general class of $f$-divergence statistical measures. We believe that this contribution has broader applicability beyond this topic.


-  The combination of using a principled measure of distance to the goal and a real world cost function (we know of only one other paper [4] that is as precise in establishing real world costs) is new in our set up.  This is essential to achieving truly optimal real-world advice
\item Finally, we introduce the verification procedure, which we argue is essential when the goal is to effect the true class probabilities. This is because research on adversarial examples has proved than ML classifiers cannot be trusted on modified data points, making some kind of authentication necessary. The two-step procedure for creating amicable perturbations, where the verifier uses both $x$ and the perturbed $\tilde{x}$ is a novel contribution of this work.

In short, we believe that our framework includes several novel and necessary ideas for the task of using ML classifiers to improve one's real world situation. The revised version of our paper will include more discussion of how our framework differs from previous set ups (such as algorithmic recourse and improvement-focused causal-recourse).

[1] Wachter, S., Mittelstadt, B., \& Russell, C. (2017). Counterfactual explanations without opening the black box: Automated decisions and the GDPR. Harv. JL \& Tech., 31, 841.

[2] Freiesleben, T. (2022). The intriguing relation between counterfactual explanations and adversarial examples. Minds and Machines, 32(1), 77-109.

[3]König, G., Freiesleben, T., & Grosse-Wentrup, M. (2023, June). Improvement-focused causal recourse (ICR). In Proceedings of the AAAI Conference on Artificial Intelligence (Vol. 37, No. 10, pp. 11847-11855).

[4] Ramakrishnan, G., Lee, Y. C., & Albarghouthi, A. (2020, April). Synthesizing action sequences for modifying model decisions. In Proceedings of the AAAI Conference on Artificial Intelligence (Vol. 34, No. 04, pp. 5462-5469).

---

### Author Response · Authors · 2023-11-18
**Clarification Regarding Trustworthiness**

There are several different concepts which are sometimes referred to as trustworthiness and robustness, and we believe this has led to some confusion regarding the focus of our paper and its relationship to previous works.  Our paper focuses on \textit{trustworthiness in the presence of adversarially vulnerable classifiers}.  Our goal is to ensure a modified data point $\tilde{\mathbf{x}}$ has true class probabilities $\tilde{\mathbf{y}}$ close to the target set ($d_\mathcal{Y}(\tilde{\mathbf{y}},T) \leq \delta$).  We do this by, first constructing $\tilde{\mathbf{x}}$ such that $d_\mathcal{Y}(M(\tilde{\mathbf{x}}),T) \leq \delta$,  then using our verification procedure to ensure $M(\tilde{\mathbf{x}})\approx \tilde{\mathbf{y}}$. This is important because research into adversarial attacks has shown that ML classifiers often have $M(\tilde{\mathbf{x}})\not\approx \tilde{\mathbf{y}}$ when $\tilde{\mathbf{x}}$ is a modified data point.  When we refer to trustworthiness in our paper, we are always referring to this concept.

Papers such as [1] and [2] use the similar term \textit{adversarial robustness} to refer to an entirely different concept.  A point $\tilde{\mathbf{x}}$ is robust if $\arg\max M(\tilde{\mathbf{x}}) = \arg\max M(\tilde{\mathbf{x}}')$ for all  $M(\tilde{\mathbf{x}}')$ satisfying $||\tilde{\mathbf{x}}-\tilde{\mathbf{x}}'||\leq \epsilon$.  Essentially, $\tilde{\mathbf{x}}$ is $\epsilon$ far away from the decision boundary of $M$.  This concept is distinct from our concept of trustworthiness: an adversarial example (which is misclassified) could be robust and a correctly classified point might not be robust.  This concept of robustness is more important when the goal is to change the classifier's decision (instead of changing the true class probabilities) where the small change in probability from $51\%$ to $49\%$  is critical.


Several recent papers such as [3] have focused on \textit{trustworthiness against non-causal correlations}.  In this concept of trustworthiness we wish to avoid changing features that correlate to the label but do not have a causal relationship with the label.  This is important but distinct form the concept of trustworthiness in our paper.  By exploiting weaknesses in the classifier, it is possible to produce an adversarial example, even when only changing causally related features.  It is also possible that a correctly classified point, $M(\tilde{\mathbf{x}})\approx\tilde{\mathbf{y}}$ (trustworthy using our definition) does not lead to the expected outcome because the features changed had no causal relationship to the label.  Adding casual constraints to our optimization problem would be promising avenue for future research.

In our revised paper we will clarify what we mean by trustworthiness and how this compares to the concepts of robustness and trustworthiness in previous works.

[1] Dominguez-Olmedo, R., Karimi, A. H., & Schölkopf, B. (2022, June). On the adversarial robustness of causal algorithmic recourse. In International Conference on Machine Learning (pp. 5324-5342). PMLR.

[2]Pawelczyk, M., Datta, T., van-den-Heuvel, J., Kasneci, G., & Lakkaraju, H. (2022). Probabilistically robust recourse: Navigating the trade-offs between costs and robustness in algorithmic recourse. arXiv preprint arXiv:2203.06768.

[3]König, G., Freiesleben, T., & Grosse-Wentrup, M. (2023, June). Improvement-focused causal recourse (ICR). In Proceedings of the AAAI Conference on Artificial Intelligence (Vol. 37, No. 10, pp. 11847-11855).

---

### Author Response · Authors · 2023-11-18
**Calrification Regarding the Verifier**

We believe that there were misunderstandings in the how the verifier works that we would like to clarify. In [1] the authors explain that adversarial examples exist because every set of training data contains random correlations that due not generalize to data outside of the training set. The model $M$ learns to recognize features tied to these non-generalizable correlations.  Adversarial attacks modify these non-generalizable  features generating points that are misclassified by $M$.  Although there is no adversary in our problem setting, we are concerned that solving the optimization problem (7)
$$
\arg\min_{\tilde{\mathbf{x}}\in\mathcal{A}(\mathbf{x})}\hspace{0.1cm} d_\mathcal{Y}(M(\tilde{\mathbf{x}}),T) +\lambda d_\mathcal{X}(\mathbf{x},\tilde{\mathbf{x}})
$$
may change these non-generalizable features (act adversarially) leading to an $\tilde{\mathbf{x}}$ that is misclassified.  (This is because adversarial attacks function by solving a very similar optimization problem.)  We propose a plan that takes advantage of the fact we know exactly what is shown to the optimization problem and the output of that optimization.

We create another classifier that has not learned the same non-generalizable features.  By passing $\tilde{\mathbf{x}}$  into both classifiers we can determine whether $\tilde{\mathbf{x}}$ is adversarial (to $M$) by examining the discrepancy between the two classifier's outputs.  This second classifier is the verifier $V$, which takes advantage of the fact we know both $\mathbf{x}$ and $\tilde{\mathbf{x}}$.
$V$ is trained with the same data as $M$, but for a fundamentally different classification problem: determining whether pairs of data points are in the same class.  Thus $V(\mathbf{x},\tilde{\mathbf{x}})\approx \mathbb{P}(\text{class}[\mathbf{x}] =  \text{class}[\tilde{\mathbf{x}}])$.  Because this is a different classification task, $V$ learns to recognize features differently than $M$ and does not learn the same non-generalizable features (although it may learn it's own set of non-generalizable features).

By using the formula $\mathbb{P}(\text{class}[\mathbf{x}] =  \text{class}[\tilde{\mathbf{x}}])\approx\sum_{i=1}^k M_i(\mathbf{x})M_i(\tilde{\mathbf{x}})$ we can  estimate the same probability using both $V$ and $M$.  If $\tilde{\mathbf{x}}$ acts adversarially we would expect $\sum_{i=1}^k M_i(\mathbf{x})M_i(\tilde{\mathbf{x}})$ to be very small while $V(\mathbf{x},\tilde{\mathbf{x}})$ is large.  If $\tilde{\mathbf{x}}$ is not adversarial we would expect similar values from both $\sum_{i=1}^k M_i(\mathbf{x})M_i(\tilde{\mathbf{x}})$ and $V(\mathbf{x},\tilde{\mathbf{x}})$.  This allows us to use
$$
\Delta (\mathbf{x},\tilde{\mathbf{x}}) = \left |V(\mathbf{x},\tilde{\mathbf{x}}) - \sum_{i=1}^k M_i(\mathbf{x})M_i(\tilde{\mathbf{x}})\right |
$$
to verify each $\tilde{\mathbf{x}}$.

Because $V$ is not a part of the optimization problem, the optimizer has no incentive to change any non-generalizable features learned by $V$ and we trust that $\tilde{\mathbf{x}}$ will not act adversarially against $V$ (although $\tilde{\mathbf{x}}$ could act adversarially against $M$ because $M$ is shown to the optimizer).

This theory is justified by the fact that our verification step is 100\% effective at detecting adversarial examples created by the Carlini-Wagner attack.  Our revised paper will clarify this reasoning.

**When a Perturbation Fails Verification**

Because we avoid showing $V$ to the optimizer we will occasionally generate $\tilde{\mathbf{x}}$ that  fail the verification step.  In page 6 of the paper we address several possible recourses when a perturbation fails the verification procedure.  First, decreasing $\lambda$ in order to put more emphasis on finding an effective change may be enough to fix the problem.  If that fails we can we can decrease the size of the target set (increase $p$ and decrease $q$) in order to further prioritize finding effective changes.  Finally, we can add a random perturbation to $\mathbf{x}$ before we run our optimizer in order to move away from the adversarial example.


[1] Ilyas, A., Santurkar, S., Tsipras, D., Engstrom, L., Tran, B., \& Madry, A. (2019). Adversarial examples are not bugs, they are features. Advances in neural information processing systems, 32.

---

### Author Response · Authors · 2023-11-18
**Summary of Rebuttal by Authors**

Dear reviewers,

We wish to thank all the reviewers for their time spent in understanding and evaluating our paper. We are very grateful for the valuable feedback.
In these comments we wish to address the concerns of the reviewers, answer questions, and clarify a few misunderstandings.   We hope this will increase the reviewer's understanding of our work and appreciation for our paper.

We begin by summarizing the reviewer's feedback starting with the strengths recognized in our paper.

1. The verification procedure is an interesting and novel idea. (f8ei, 6MjG, cADE)
2. The problem set up is clear and well defined. (syxy, 6MjG, cADE)
3. The paper tackles an important problem with clear applications. (syxy, v88R, cADE)
4. The mathematical analysis of the reward function is interesting and useful (sxyx, 6MjG)
5. The problem framework is novel and distinct from prior works. (sxyx)
6. The paper is well written and easy to understand. (v88R)


We now address the most common concerns amongst the reviewers.  Each of theses concerns is dealt with in greater detail in its own comment below.

1. **Clarification Regarding Verifier**: Some reviewers wanted more analysis/intuition behind how the verifier works.  In a comment below we clarify the motivation behind the verifier's design and how the goal of the verifier within this context is different from conventional adversarial defense.

2. **Clarification Regarding Framework Novelty**: Some reviewers had questions regarding the novelty of our framework.  We note that the task of changing true class probabilities is very different from the task of merely changing classifier outputs (which is the focus of most related works on counterfactuals and algorithmic recourse).  We have found one previous work [2] which clearly highlighted this difference and focused on the task of changing true class probabilities.  We believe, however, that the concepts of (1) flexible target sets, (2) principled measures of distance to the target (and a theoretical analysis thereof), and (3) a verification procedure (all novel in our paper) are necessary to effectively solve this problem.  The introduction of these concepts makes our paper the first to fully develop the task of changing true class probabilities.  We elaborate on the distinction between our paper and prior works in a comment below.

3. **Clarification Regarding Trustworthiness**: There was some confusion between the concept of trustworthiness presented by our paper (which focuses on how a classifier might misclassify a modified input) and the separate concepts of causality (v88R, 6MjG) and adversarial robustness (f8ei).  In a comment below we highlight the differences between all of these important concepts and the novelty of what our paper seeks to achieve.


We hope that this summary and the clarification found in our comments will ease the reviewer's concerns and enhance their opinion of our paper.  Answers to specific questions from each reviewer may be found in comments to the individual reviews.  We respectfully request that the reviewers update their evaluations of our paper in light of these clarifications and welcome additional requests for clarification.  We would like to thank the reviewers again for their feedback.


Best regards,


The Authors

---

### Author Response · Authors · 2023-11-22
**Comment to Reviewers**

Dear Reviewers,

We would like to thank you again for your time and effort spent in reviewing our paper and providing constructive criticism.  As the time for reviewer/author discussion comes to a close, we request that you submit any comments or questions regarding our clarifications as soon as possible.  We hope that any updates to our paper's ratings or confidence scores will also be posted soon.

Best regards,

The Authors

---

### Meta-Review · Area_Chair_CafV · 2023-12-09

**Metareview:**

**Summary**
This paper introduces a post-hoc model evaluation framework to find "amicable perturbations" of features to achieve a desired outcome. The work introduces an optimization-based approach to output amicable perturbations -- which combines a generation step and a verification step. The authors evaluate their approach on tabular datasets, finding that they can achieve perturbations that achieve lower cost than their baselines.

**Strengths**

- Scope: The paper studies a timely topic that has important applications.
- Clarity: The paper is well-written and easy to understand.
- Technical Novelty: The paper includes several interesting technical contributions (e.g., a "verification step"). This is noteworthy given the saturation of methods in this area.

**Weaknesses and Suggestions**

- Soundness: The proposed method suffers from critical modes that change based on its use case. If the goal is to use "amicable perturbations" as an internal auditing tool (e.g., for adversarial robustness), then it should output a certificate of robustness, and the authors should describe how practitioners can set the cost function to the model to get a meaningful guarantee of robustness. If the goal is to present "amicable perturbations" to decision subjects (e.g., recourse provision), then the method should support hard actionability constraints, and the paper should include a description of how practitioners should handle settings for when recourse does not exist.

-Significance/Positioning: The paper casts "amicable perturbations" as a new framework -- a claim that received substantial criticism from reviewers. Having read through the paper, review, and rebuttal - I agree with Reviewers v88R and f8ei that seems like an oversell. I want to be clear that there is value in finding perturbations that alter the true outcome in established frameworks - namely, recourse, counterfactual explanations ($\neq$ recourse), counterfactual invariance, and adversarial robustness. I would recommend the authors choose to focus on this aspect of the work and highlight its value in one of the frameworks described above. If the goal is recourse, as it appears to be, then the authors should familiarize themselves with [2].

----

[1] Ustun et al (2019). "Actionable Recourse in Linear Classification"
[2] Freiesleben et al (2023), “Improvement-Focused Causal Recourse"

**Justification For Why Not Higher Score:**

See review.

**Justification For Why Not Lower Score:**

N/A

---

### Decision · Program_Chairs · 2024-01-16

Reject